# Integrin CD103 reveals a distinct developmental pathway of autoreactive thymocytes in TCR transgenic mice

Nurcin Liman [1], Can Li [1], Megan A. Luckey[1], Hilary R. Keller [1], Jie Li[1], William Hajjar[1], Jan Wisniewski[1], Michael Kruhlak [2], Vanja Lazarevic [1] & Jung-Hyun Park [1] ✉

Clonal deletion through negative selection is critical to eliminate autoreactive T cells in the thymus. Negative selection, however, is imperfect such that some autoreactive thymocytes can escape thymic deletion and successfully populate peripheral tissues. This is also the case for autoreactive 2D2 TCR transgenic T cells, a widely employed mouse model in studying the pathogenesis of CD4 T cell-mediated experimental autoimmune encephalomyelitis. How autoreactive 2D2 thymocytes evade negative selection, however, remains incompletely understood. Here we show that negative selection of MHC-II-restricted thymocytes, specifically 2D2 TCR transgenic T cells, is associated with the induction of integrin CD103, and that forced expression of CD103 downregulates CXCR4 expression, alters intra-thymic trafficking, and reinforces clonal deletion of immature thymocytes. Stratification of positively versus negatively selected 2D2 T cells based on their distinct coreceptor expression further shows that CD103 does not affect the generation of conventional CD4 T cells but is deleterious for autoreactive CD4, CD8 double-negative 2D2 T cells that correspond to CD69-negative CCR7-intermediate thymocytes, displaying markers of agonistic TCR signalling. Collectively, these results propose CD103 expression as an indicator and contributor of negative selection for MHC-II-restricted T cells, providing further mechanistic insights into the process of T cell selection in the thymus.

The generation, lineage choice, and functional differentiation of T cells critically depend on the antigen specificity of their T cell receptors (TCR)[1]. In agreement, the transgenic expression of a clonotypic TCR is sufficient to impose MHC restriction and antigen specificity onto developing thymocytes, such that TCR transgenic T cells replicate the lineage and antigen reactivity of the T cell clone of which the recombined TCR genes have been isolated from[2]. The TCR also mediates negative selection in the thymus[3]. Accordingly, transgenic mice expressing self-antigen-specific TCRs fail to generate functionally mature T cells as they undergo thymic clonal deletion[4]. In this regard,

the 2D2 TCR transgenic mice are interesting because their TCR is specific to the myelin oligodendrocyte glycoprotein (MOG), a self-antigen that is expressed in the myelin sheath[5,6], but CD4 T cells carrying this TCR transgene somehow bypass thymic negative selection[5]. Remarkably, 2D2 T cells are not only spared from clonal deletion but also proliferate extensively, representing 95% of the mature peripheral T cell pool in 2D2 TCR transgenic mice[7]. How 2D2 TCR transgenic thymocytes undergo positive selection but can escape negative selection is not fully understood. Possibly, the agonistic MOG self-peptides recognized by the 2D2 TCR are not present in the thymus or

[1]Experimental Immunology Branch, Center for Cancer Research, National Cancer Institute, NIH, Bethesda, MD, USA. [2]Laboratory of Cancer Biology and Genetics, Center for Cancer Research, National Cancer Institute, NIH, Bethesda, MD, USA. ✉e-mail: Parkhy@mail.nih.gov

are not abundant enough to mediate the negative selection of 2D2 TCR thymocytes[5]. However, experimental evidence supporting either of these explanations is not fully available. Thus, mechanistic insights into the thymic development and differentiation of the 2D2 TCR transgenic T cells are lacking, and how the thymic selection processes of 2D2 CD4 T cells differ from that of conventional MHC-II restricted CD4 T cells has not been addressed.

Because of its MOG-specific reactivity, a major pathological target in multiple sclerosis (MS)[8], the 2D2 TCR and transgenic mice expressing this receptor (2D2 mice) have been frequently used in experimental autoimmune encephalomyelitis (EAE), an animal model of MS[9–12]. Along these lines, CD4 T cells from 2D2 mice are used in passive transfer models of EAE[9], and 2D2 mice themselves are also heavily utilized in autoimmunity research[5]. Given the critical role of 2D2 TCR transgenic CD4 T cells in initiating and driving neuropathology, it would be important to delineate their developmental trajectory in greater detail. Thus, here, we wished to unravel how self-reactive CD4 T cells that escape negative selection differ in their thymic development from conventional CD4 T cells that are specific to non-self-antigens. To this end, we employed other MHC-II-restricted TCR transgenic mouse models, such as the moth cytochrome C (MCC)-specific AND TCR[13], and the ovalbumin (OVA)-specific OT-II TCR[14], to assess their thymic differentiation in comparison to 2D2 mice. To further correlate the positive versus negative selection of autoreactive 2D2 T cells with changes in differentiation marker expression, we also surveyed the expression of a series of surface proteins during the thymic development of 2D2 thymocytes.

As a result, here we identify the cell retention molecule CD103, also known as integrin $\alpha_E$[15], as a prominent marker for 2D2 T cells receiving agonistic TCR signaling and undergoing negative selection. CD103 is an integrin that binds to E-cadherin[16], and is typically induced only on MHC-I- and not MHC-II-signaled post-selection thymocytes[17]. Importantly, CD103 expression promotes the clonal deletion of 2D2 thymocytes because forcing CD103 expression selectively interferes with the early differentiation of 2D2 thymocytes. Collectively, these results describe an unexpected role for CD103 in the thymic selection and differentiation of MHC-II-restricted autoreactive T cells, further revealing the importance of thymocyte interaction with E-cadherin-expressing thymic epithelial cells (TEC).

## Results

### Thymic development of 2D2 TCR transgenic T cells

We embarked on this study to understand the developmental trajectory of autoreactive T cells in the thymus, employing the 2D2 TCR transgene as an experimental model. Assessing the CD4 versus CD8 profiles of 2D2 thymocytes showed that CD4 single-positive (SP) T cell generation in 2D2 mice was significantly increased compared to wild-type (WT) mice, concordant with the 2D2 TCR being restricted to MHC-II (Fig. 1A)[5]. Examination of surface markers that are associated with thymocyte selection and maturation, i.e., CD5, CCR7, CD24, MHC-I, and CD69[18,19], indicated that 2D2 CD4SP cells were comparable to WT CD4 T cells in their differentiation status (Suppl. Fig. 1A). However, we also noticed that CD5 and CD69, whose expression is induced upon TCR signaling, were slightly decreased on CD4SP cells from 2D2 mice compared to WT mice, suggesting that the 2D2 TCR can mediate positive selection with weaker TCR signaling. Increased positive selection of CD4 T cells is common to MHC-II-specific TCR transgenic thymocytes as illustrated in AND TCR transgenic mice whose TCR is not autoreactive but specific to MCC peptides in the context of MHC-II (Fig. 1A)[13]. Notably, the frequency of immature CD4, CD8 double-positive (DP) thymocytes did not differ between 2D2 and AND transgenic mice (Suppl. Fig. 1B), despite 2D2 DP cells expressing an autoreactive TCR (Suppl. Fig. 1C). Thus, the generation of CD4 T cells in 2D2 TCR transgenic mice appeared to be intact despite the autoreactivity of their TCR.

To further assess whether and how autoreactive 2D2 TCRs differ from the non-autoreactive AND TCRs in their thymic development, we next determined total thymocyte numbers in these mice. The forced and premature expression of TCR transgenes disturbs early thymopoiesis[20]. In agreement, total thymocyte numbers were significantly decreased in both AND and 2D2 mice (Fig. 1B). Unlike AND thymocytes, however, 2D2 thymocytes showed a substantial increase in the frequency and number of CD4, CD8 double-negative (DN) cells compared to both WT and AND mice (Fig. 1C). Contrary to AND DN thymocytes, however, most 2D2 DN cells displayed high levels of TCRβ (Fig. 1D), expressing the 2D2 clonotypic TCRs, i.e., Vα3.2 and Vβ11 (Fig. 1E)[5]. Thus, the increased frequency of 2D2 DN thymocytes is associated with an increase in 2D2 TCRβ-expressing DN cells. To examine if such accumulation of TCRβ^hi DN cells is specific to the 2D2, we gated on TCRβ^hi cells and assessed the CD4 versus CD8 profiles of WT, AND, and 2D2 thymocytes (Fig. 1F). Indeed, 2D2 thymocytes were highly enriched for TCRβ^hi DN cells (Fig. 1F, G), and these 2D2 DN cells corresponded to post-selection thymocytes as demonstrated by their increased expression of MHC-I and the chemokine receptor CCR7, which are highly induced upon positive selection (Fig. 1H)[21]. These results indicate that the autoreactive 2D2 TCR promotes the generation of mature TCRβ^hi DN T cells, which are presumably the products of negative selection induced by agonistic TCR signaling[22,23]. The CD4SP T cells in 2D2 mice, on the other hand, would correspond to CD4 T cells that underwent normal thymic maturation appropriate for MHC-II-restricted cells. Thus, in 2D2 mice, CD4SP cells are the results of positive selection while DN thymocytes are likely the products of negative selection.

To understand how and when the differentiation of negatively selected 2D2 thymocytes diverges from that of positively selected cells, we focused on TCRβ^hi DP cells, which are the immediate progenies of positive selection and the precursors of both mature DN and CD4SP cells. Here we found that 2D2 mice contained significantly more TCRβ^hi DP cells than AND mice (Fig. 1G). Concurrently, there was a marked reduction in the frequency of TCRβ^hi CD4SP cells alongside an increase in TCRβ^hi DN cells (Fig. 1G). These results suggested a lineage bifurcation of TCRβ^hi DP thymocytes into TCRβ^hi DN versus CD4SP cells that is determined by the outcome of thymic selection.

To explore how the autoreactivity of 2D2 thymocytes gives rise to two distinct mature T cell populations, we next assessed the expression of a series of surface markers that are associated with thymocyte selection and differentiation, including γc family cytokine receptors and TCR-induced activation molecules (Suppl. Fig. 1D, 1E). Specifically, we focused on molecules that are absent in positively selected TCRβ^hi CD4SP cells but induced in negatively selected TCRβ^hi DN cells. Here we identified a distinct pattern of integrin expression in CD4SP and DN thymocytes, whereby the tissue retention molecules CD103, also known as integrin $\alpha_E$[24], was highly induced on DN but not on CD4SP and DP cells (Fig. 1I, top). CD103 pairs with integrin β7, which was also highly expressed on DN cells (Fig. 1I, bottom). On the other hand, the expression of integrin α4, which can pair with both β7[25] and integrin β1, remained unaltered in 2D2 DN thymocytes (Fig. 1J). Collectively, these results suggest that CD103 expression on 2D2 DN thymocytes is associated with the negative selection of an autoreactive TCR.

### Forced expression of CD103 promotes the accumulation of mature 2D2 DN thymocytes

Since CD103 was induced on negatively selected 2D2 DN thymocytes, we next asked whether forcing CD103 expression could enhance negative selection. To this end, we crossed the 2D2 TCR transgene with CD103 and integrin β7 double transgenic mice (CD103^Tgβ7^Tg)[17] to generate 2D2.CD103^Tgβ7^Tg mice. We previously reported that the combined overexpression of CD103 and integrin β7, but not either of them alone, can dramatically increase the surface expression of CD103 and β7 on immature DP thymocytes (Fig. 2A, top, Suppl. Fig. 2A)[17]. Our

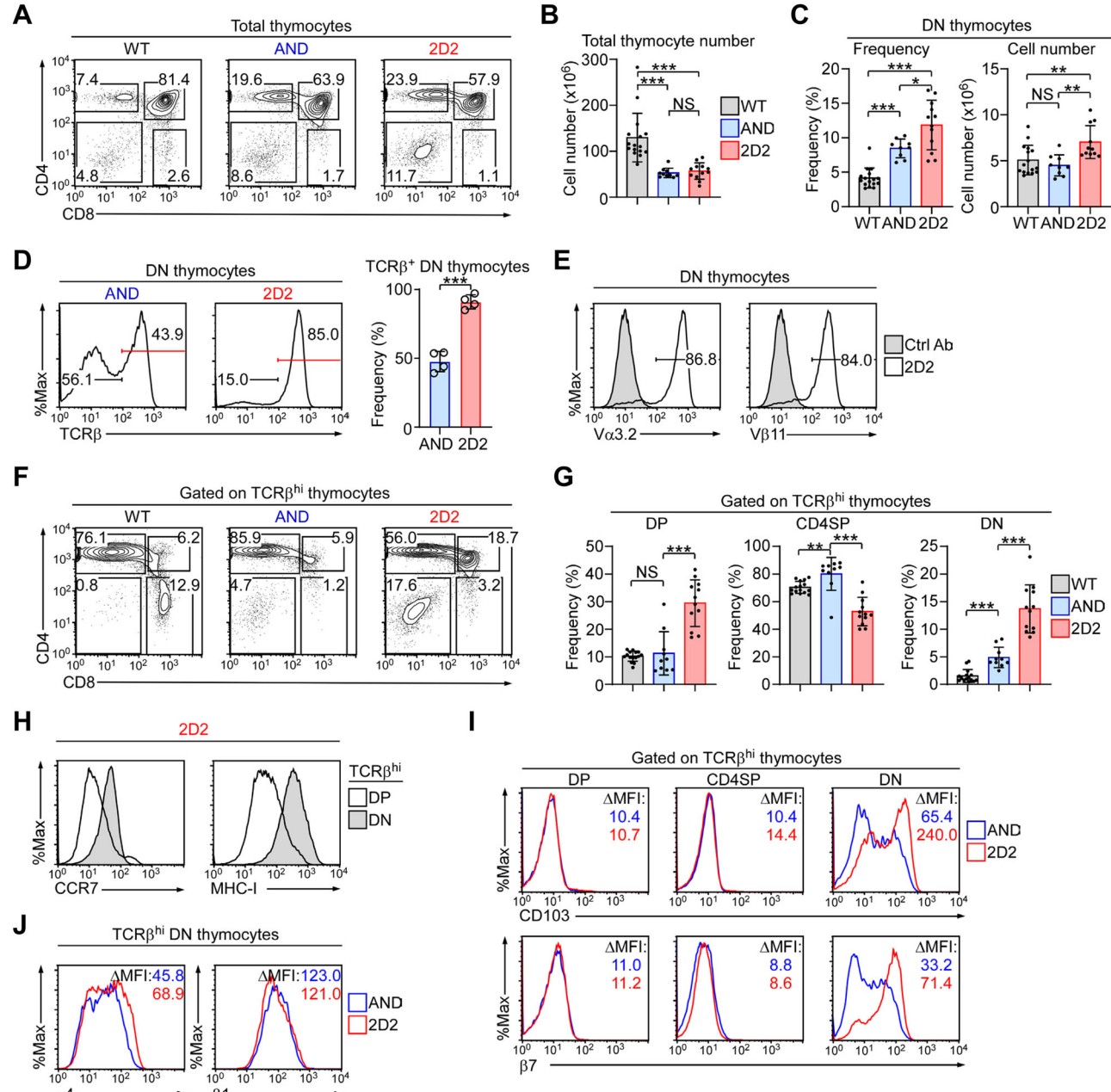

**Fig. 1 | Dysregulated thymopoiesis of 2D2 TCR transgenic mice. A** Contour plots show the CD4 versus CD8 profile of total thymocytes in WT, AND, and 2D2 transgenic mice. The results are representative of at least 6 independent experiments. **B** Bar graphs summarize the total thymocyte numbers of WT ($n = 16$), AND ($n = 10$), and 2D2 ($n = 12$) transgenic mice from 11 independent experiments. Unpaired two-tailed Student's $t$-test was used to calculate $P$ values. ***, $P < 0.001$; NS, not significant. One-Way Anova $P < 0.0001$. Data are presented as mean values ± Standard Deviation (SD). **C** Bar graphs show the frequencies and numbers of DN thymocytes in WT ($n = 16$), AND ($n = 9$), and 2D2 ($n = 12$) transgenic mice as summarized from 11 independent experiments. Unpaired two-tailed Student's $t$-test was used to calculate $P$ values. *, $P < 0.05$; **, $P < 0.01$; and ***, $P < 0.001$; NS, not significant. DN thymocytes frequency One-Way Anova $P < 0.0001$; DN thymocytes cell number One-Way Anova $P = 0.0071$. Data are presented as mean values ± SD. **D** Histograms show TCRβ expression on the DN thymocytes of AND (left) and 2D2 (right) transgenic mice. The bar graph summarizes the frequency of TCRβ⁺ DN thymocytes with 4 mice per group. Unpaired two-tailed Student's $t$-test was used to calculate $P$ values. ***, $P < 0.001$. Data are presented as mean values ± SD. **E** Histograms show expression of the clonotypic TCRα Vα3.2 (left) and TCRβ Vβ11 (right) expression

(black) on DN thymocytes in 2D2 transgenic mice. The results are representative of 3 independent experiments. Ctrl Ab: control antibody. **F** Contour plots show the CD4 versus CD8 profiles of TCRβhi mature thymocytes from WT, AND, and 2D2 transgenic mice. Results are representative of 6 independent experiments. **G** Bar graphs show the frequencies of TCRβhi DP, CD4SP, and DN thymocytes from WT ($n = 15$), AND ($n = 10$), and 2D2 ($n = 12$) transgenic mice as summarized from 11 independent experiments. Unpaired two-tailed Student's $t$-test was used to calculate $P$ values. **, $P < 0.01$; and ***, $P < 0.001$; NS, not significant. TCRβhi DP One-Way Anova $P < 0.0001$; TCRβhi CD4SP One-Way Anova $P < 0.0001$; TCRβhi DN One-Way Anova $P < 0.0001$. Data are presented as mean values ± SD. **H** Histograms show CCR7 and MHC-I (H-2KbDb) expression on TCRβhi DP versus TCRβhi DN thymocytes of 2D2 transgenic mice. Results are representative of at least 3 independent experiments. **I** Histograms show CD103 and β7 expression on TCRβhi-gated DP, CD4SP, and DN thymocytes of AND and 2D2 transgenic mice. Results are representative of at least 7 independent experiments. **J** Histograms show integrin α4 and β1 expression on TCRβhi DN thymocytes of AND and 2D2 transgenic mice. Results are representative of at least 3 independent experiments.

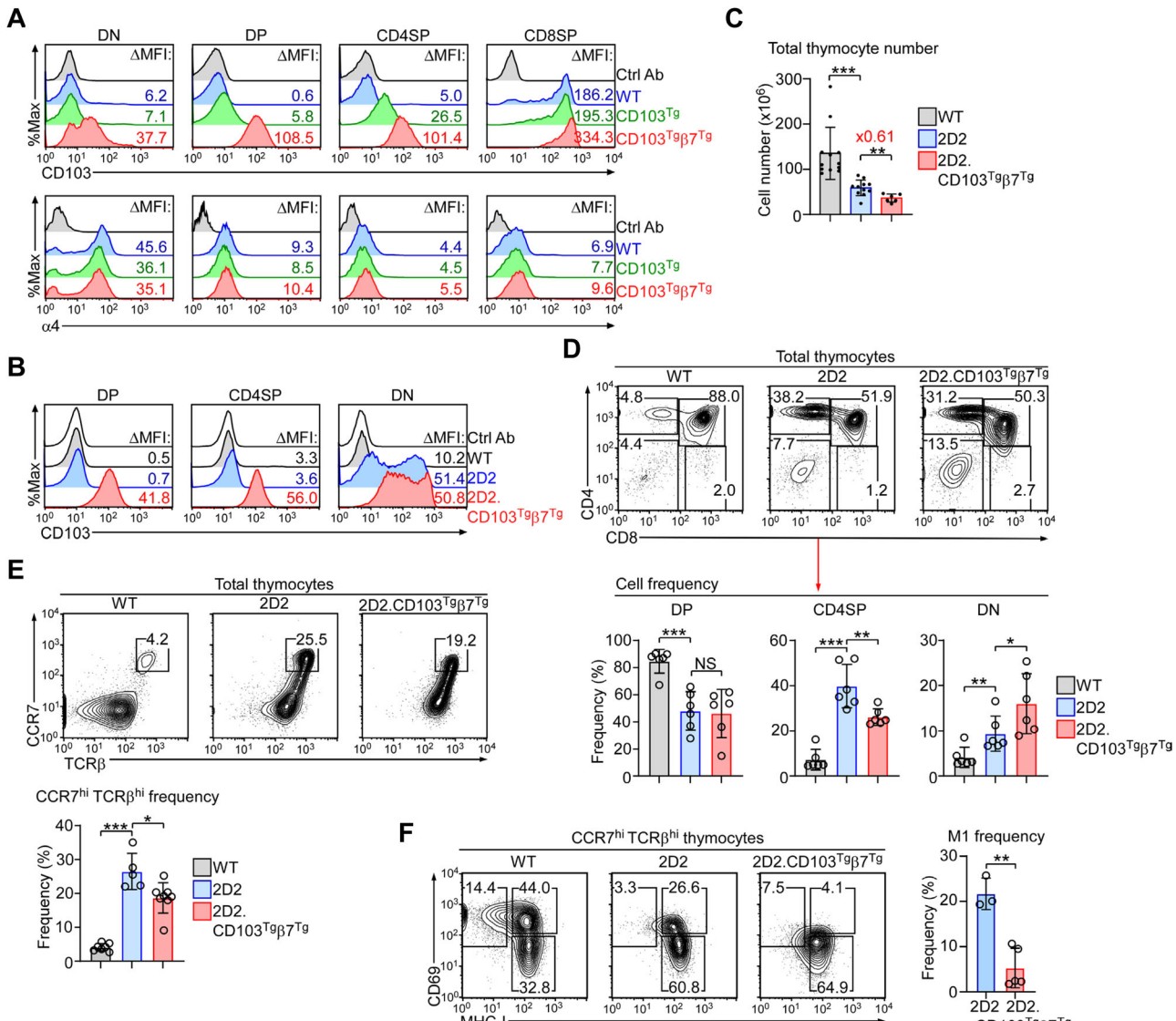

**Fig. 2 | CD103 overexpression impairs thymic development of 2D2 transgenic mice. A** Cell surface expression of integrin CD103 (top) and α4 (bottom) in different thymocyte populations of WT, CD103$^{Tg}$, and CD103$^{Tg}$β7$^{Tg}$ mice. Histograms are representative of at least 10 independent experiments. Ctrl Ab: control antibody. **B** Histograms show CD103 expression on the DP, CD4SP, and DN thymocytes of WT, 2D2, and 2D2.CD103$^{Tg}$β7$^{Tg}$ mice. Results are representative of at least 3 independent experiments. Ctrl Ab: control antibody. **C** Bar graph summarizes the total thymocyte numbers in WT ($n$ = 12), 2D2 ($n$ = 11), and 2D2.CD103$^{Tg}$β7$^{Tg}$ ($n$ = 6) mice. Overexpression of CD103 and β7 results in 0.61-fold decrease in thymocyte numbers in 2D2 mice. Results are a summary of 6 independent experiments. Unpaired two-tailed Student's $t$-test was used to calculate $P$ values. **, $P$ < 0.01; and ***, $P$ < 0.001. One-Way Anova $P$ < 0.0001. Data are presented as mean values ± SD. **D** Contour plots (top) show CD4 versus CD8 profiles of WT, 2D2, and 2D2.CD103$^{Tg}$β7$^{Tg}$ thymocytes. Bar graph summarizes the frequencies of DP, CD4SP, and DN cells among total thymocytes of WT, 2D2, and 2D2.CD103$^{Tg}$β7$^{Tg}$ mice (bottom). Contour plots are representative and bar graphs show the summary of 5 independent experiments with a total of 6 mice per group.

Paired two-tailed Student's $t$-test was used to calculate $P$ values. *, $P$ < 0.05; **, $P$ < 0.01, and ***, $P$ < 0.001; NS, not significant. DP One-Way Anova $P$ = 0.0003; CD4SP One-Way Anova $P$ < 0.0001; DN One-Way Anova $P$ = 0.0018. Data are presented as mean values ± SD. **E** Contour plots identify positively selected cells among total thymocytes by their CCR7 versus TCRβ expression (top). Bar graph summarizes the frequencies of CCR7$^{hi}$TCRβ$^{hi}$ thymocytes of WT ($n$ = 7), 2D2 ($n$ = 5), and 2D2.CD103$^{Tg}$β7$^{Tg}$ ($n$ = 8) mice (bottom). Contour plots are representative, and the bar graph shows the summary of 3 independent experiments. Unpaired two-tailed Student's $t$-test was used to calculate $P$ values. *, $P$ < 0.05, ***, $P$ < 0.001. One-Way Anova $P$ < 0.0001. Data are presented as mean values ± SD. **F** Contour plots show CD69 versus MHC-I expression of CCR7$^{hi}$TCRβ$^{hi}$ thymocytes from WT, 2D2, and 2D2.CD103$^{Tg}$β7$^{Tg}$ mice. Based on the differential expression of CD69 and MHC-I, three distinct populations are identified and their frequencies are shown. Contour plots are representative of 3 independent experiments. Bar graph summarizes the frequency of M1 population in WT, 2D2 ($n$ = 3), and 2D2.CD103$^{Tg}$β7$^{Tg}$ ($n$ = 5) mice. Unpaired two-tailed Student's $t$-test was used to calculate $P$ values. **, $P$ < 0.01. Data are presented as mean values ± SD.

further analyses now revealed that such increased expression was due to an increase in total (both cytosolic and surface) protein expression (Suppl. Fig. 2B), and that it is specific to CD103 but not α4 (Fig. 2A, bottom), the only other integrin α-chain that pairs with β7[26]. Thus, we considered 2D2.CD103$^{Tg}$β7$^{Tg}$ mice as an appropriate model to examine the effect of increased CD103 expression on thymocyte selection in vivo.

As expected, CD103 levels in 2D2.CD103$^{Tg}$β7$^{Tg}$ mice were upregulated across all thymocyte subsets (Fig. 2B), and such an increase in CD103 levels was further associated with a deleterious effect on thymopoiesis (Fig. 2C). Notably, the total thymocyte numbers of 2D2.CD103$^{Tg}$β7$^{Tg}$ mice were 0.61-fold decreased compared to 2D2 mice (Fig. 2C), which was accompanied by a significant increase in DN but a marked decrease in CD4SP cell frequencies (Fig. 2D). These

results suggest that the forced expression of CD103 promotes negative selection (reflected in the increase of DN cells) but impairs positive selection (observed by the loss of CD4SP cells) in 2D2 mice. In agreement, positively selected thymocytes, as identified by high-level expression of both CCR7 and TCRβ (CCR7hiTCRβhi) (Fig. 2E, top)[21,27], were significantly reduced in 2D2.CD103Tgβ7Tg mice compared to control 2D2 mice (Fig. 2E, bottom). Post-selection CCR7hiTCRβhi thymocytes can be further subdivided into three distinct stages based on their CD69 versus MHC-I expression[28]. Semi-mature (SM) cells, identified as CD69+MHC-I−, further differentiate into Mature 1 (M1; CD69+MHC-I+) and then Mature 2 (M2; CD69−MHC-I+) cells during their terminal maturation (Suppl. Fig. 3A). In 2D2 mice, the overall frequencies of SM and M1 cells are significantly decreased compared to WT mice (Suppl. Fig. 3B), suggesting that thymic selection is suboptimal in 2D2 mice. In this regard, we documented that the transgenic 2D2 TCR was prematurely expressed at the earliest stage of immature DN thymocytes (Suppl. Fig. 3C, D), which could have caused less efficient thymic selection of 2D2 cells. Notably, CD103 overexpression exacerbates these effects as CCR7hiTCRβhi cells in 2D2.CD103Tgβ7Tg mice were mostly comprised of end-differentiated M2 stage (CD69lowMHC-Ihi) thymocytes and lacked M1 stage (CD69+MHC-I+) cells (Fig. 2F). Altogether, these results indicate that the production of mature thymocytes is significantly impaired by the overexpression of CD103.

Because positive selection is impaired, we conversely expected that negative selection should be enhanced in 2D2.CD103Tgβ7Tg mice. To address this point, we employed the chemokine receptors CXCR4 and CCR7 to assess thymic selection independently of transgenic TCR expression (Suppl. Fig. 3E). CXCR4 is highly expressed on pre-selection thymocytes but downregulated upon their maturation[29,30], while CCR7 is absent on immature thymocytes but induced upon TCR signalling[31]. By assessing these markers, we identified negatively selected cells as CXCR4-negative CCR7-intermediate (CXCR4−CCR7int) thymocytes (Fig. 3A, red boxes), whose frequencies were highly increased in 2D2 mice and even more so in 2D2.CD103Tgβ7Tg mice (Fig. 3A). These results are concordant with our hypothesis that 2D2 and 2D2.CD103Tgβ7Tg mice contain large numbers of negatively selected thymocytes. Moreover, CXCR4−CCR7int thymocytes were highly enriched for DN cells that correspond to negatively selected mature thymocytes, based on their loss of the selection marker CD69 (Fig. 3B). Of note, we realized that the CXCR4−CCR7int population contained a significant fraction of CD69+ DP cells, which are presumably immature thymocytes in the process of undergoing positive selection (Fig. 3B). Thus, we considered it important to refine the phenotype of negatively selected thymocytes using alternative markers. In this regard, thymocyte maturation can be visualized by the differential expression of CD69 versus CCR7[29], identifying 6 distinct populations that we refer to henceforth as population I – population VI (Suppl. Fig. 3F). When applying this plotting strategy to WT, 2D2, and 2D2.CD103Tgβ7Tg thymocytes, we found a dramatic and gradual increase in CD69−CCR7int cells, i.e., population VI (Fig. 3C), supporting our notion that negative selection in 2D2 mice is exacerbated by CD103 overexpression (Fig. 3C). Indeed, gating on population VI cells turned out to be an accurate tool to identify negatively selected cells as the vast majority were DN cells (Fig. 3D). Population VI DN cells expressed high levels of CD103 (Fig. 3E, top), concomitant to large amounts of Helios and PD-1, which are associated with agonistic TCR signaling and negative selection (Fig. 3E, bottom, 3F)[32,33]. In marked contrast, population V CD4SP cells that correspond to positively selected CD4SP thymocytes did not express any of these markers (Fig. 3E, F). Thus, population VI DN cells are negatively selected cells, which we also affirmed by their increased active caspase-3 expression (Fig. 3G).

Finally, to test the role of CD103 in this process, we obtained CD103-deficient mice (i.e., Itgae−/−) to examine whether the lack of CD103 would interfere with negative selection. When assessing the

frequency of population VI cells in mature thymocytes of WT versus Itgae−/− mice (Suppl. Fig. 4A), we found that CD103 deficiency significantly impaired negative selection as illustrated by the marked decrease in population VI cells (Fig. 3H). Thus, we identified CD69−CCR7int (population VI) cells as the product of thymic negative selection. Our results further indicate that integrin CD103 would promote the negative selection of autoreactive thymocytes while impeding their positive selection.

## CD103 overexpression impairs thymic development of MHC-II-specific T cells

To test the effects of CD103 on non-autoreactive TCR transgenes, we next crossed the CD103Tgβ7Tg with AND TCR transgenic mice[13]. Akin to 2D2.CD103Tgβ7Tg thymocytes, AND.CD103Tgβ7Tg thymocytes contained increased frequencies of DN cells compared to AND mice (Fig. 4A), which was associated with a decrease in total thymocyte numbers (Fig. 4A, bottom). To associate the reduced thymocyte numbers with increased negative selection and diminished positive selection, we assessed the CXCR4 versus CCR7 profile of AND and AND.CD103Tgβ7Tg thymocytes (Fig. 4B)[29]. Consistent with enhanced negative selection by CD103 overexpression, AND.CD103Tgβ7Tg thymocytes contained increased frequencies of CXCR4−CCR7int cells that were enriched for DN thymocytes (Fig. 4B). Similarly, CD103 overexpression promoted negative selection in other non-autoreactive TCR transgenes as well which we documented with OT-II TCR transgenic mice[14]. As expected, DN cell frequencies were substantially increased while overall thymocyte numbers were significantly decreased in OT-II.CD103Tgβ7Tg mice (Fig. 4C). Akin to AND.CD103Tgβ7Tg thymocytes, a significant fraction of CXCR4−CCR7int cells in OT-II.CD103Tgβ7Tg mice were also DN cells (Fig. 4D). To confirm that this population is a product of negative selection, we assessed PD-1 and Helios expression in population VI cells (Fig. 4E), which were highly induced in both OT-II and OT-II.CD103Tgβ7Tg population VI cells (Fig. 4F). Altogether, these results showed that CD103 overexpression profoundly alters the development of thymocytes, and they further suggested that the clonal deletion of autoreactive thymocytes can be reinforced by CD103-mediated mechanisms.

Lastly, we asked whether CD103 expression can be established as a general marker of thymic negative selection. To this end, we employed the classical autoreactive mouse model of the HY TCR transgene[34]. The HY TCR is specific to a male antigen such that HY TCR thymocytes in male mice are agonistically signaled to undergo negative selection[34]. Interestingly, when plotting HY male thymocytes for CD69 and CCR7 expression, we found that most of the thymocytes corresponded to population VI cells that were phenotypically equivalent to negatively selected 2D2 DN thymocytes (Fig. 4G). In fact, population VI cells were highly increased in autoreactive HY TCR male thymocytes (Fig. 4H), and they expressed large amounts of PD-1 (Fig. 4I, right), confirming them as products of negative selection. Because population VI HY cells also expressed high levels of CD103 (Fig. 4I, left), these results support CD103 expression as a marker of negative selection in DN thymocytes.

## Dysregulated CXCR4 expression in TCR transgenic thymocytes
Negative selection in the thymus is mediated by TECs, primarily by medullary TECs (mTEC)[35]. Incidentally, TECs express large but variable amounts of E-cadherin (Fig. 5A)[36], and our reanalyses of public mTEC single cell RNA-sequencing (scRNA-Seq) data revealed a robust and broad expression of Cdh1, encoding E-cadherin, among different mTEC populations (Fig. 5B). Notably, Cdh1 transcripts were present in Aire+ mTECs, CCL21-producing mTECs, and thymic Tuft cells (Trpm5+), indicating that E-cadherin expression is possibly a feature associated with thymic stromal cells (Fig. 5B). In this regard, we found it interesting that none of the peripheral lymph node stromal cells express E-cadherin (Suppl. Fig. 4B).

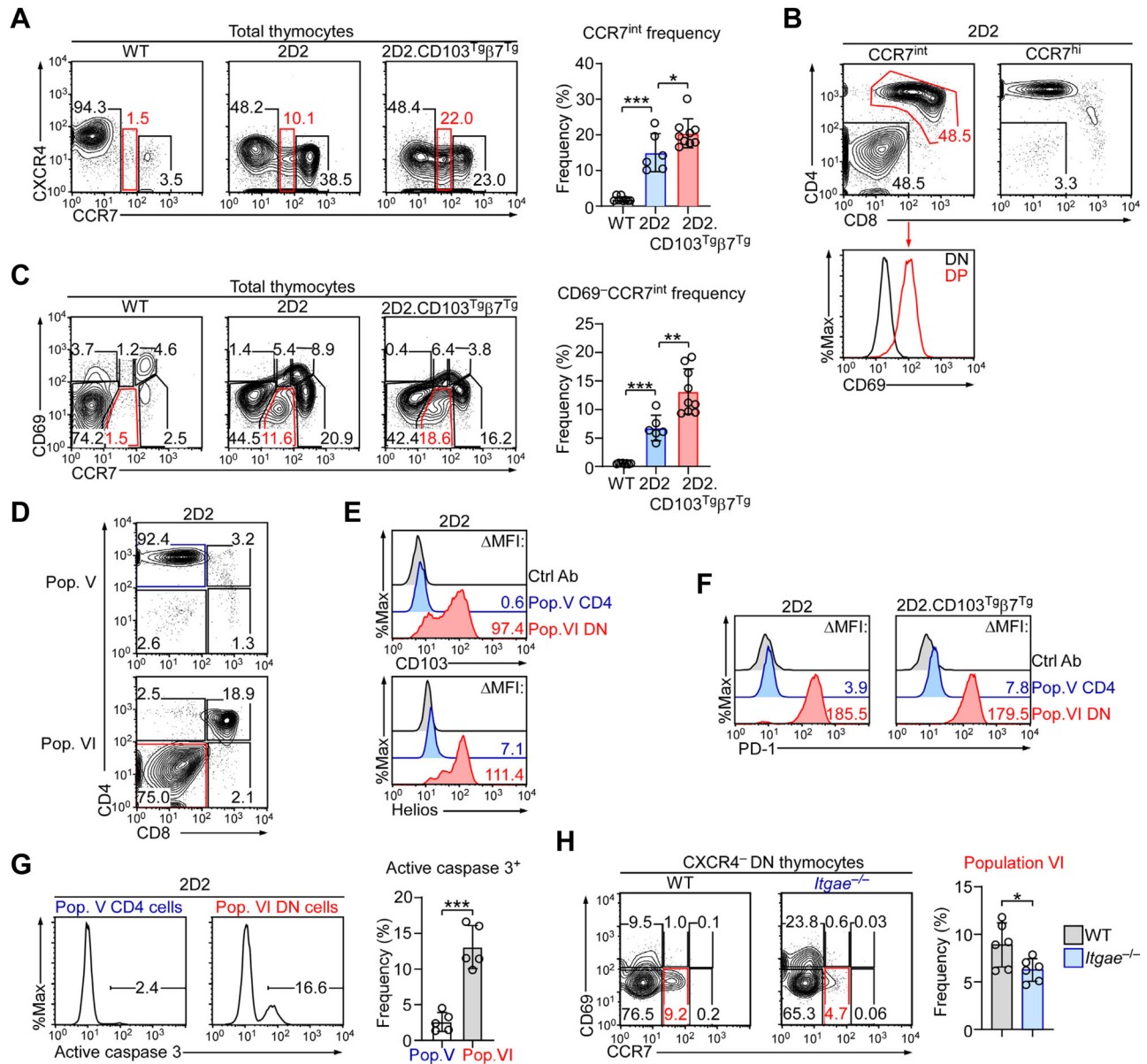

**Fig. 3 | CD103 overexpression promotes negative selection in 2D2 transgenic mice. A** Contour plots show the CXCR4 versus CCR7 profiles of WT ($n = 8$), 2D2 ($n = 6$), and 2D2.CD103$^{Tg}$β7$^{Tg}$ ($n = 9$) thymocytes, identifying three distinct populations based on their CCR7 levels, *i.e.*, CCR7$^{neg}$, CCR7$^{int}$, and CCR7$^{hi}$. Bar graphs (right) show the frequencies of CCR7$^{int}$ thymocytes (boxed in red, left) of the corresponding mice. Contour plots are representative, and the bar graph shows the summary of 4 independent experiments. Unpaired two-tailed Student's *t*-test was used to calculate *P* values. *, $P < 0.05$, ***, $P < 0.001$. One-Way Anova $P < 0.0001$. Data are presented as mean values ± SD. **B** Contour plots (top) show CD4 versus CD8 profiles of CCR7$^{hi}$ and CCR7$^{int}$ thymocytes of 2D2 mice. Histogram (bottom) shows CD69 expression on DN (black) and DP (red) cells of CCR7$^{int}$-gated 2D2 thymocytes. Contour plots are representative of 4 independent experiments. **C** Contour plots show the CD69 versus CCR7 profiles of WT ($n = 8$), 2D2 ($n = 6$), and 2D2.CD103$^{Tg}$β7$^{Tg}$ ($n = 9$) thymocytes that identify six distinct populations based on their differential expression of CD69 and CCR7. Starting from CD69, CCR7 double-negative cells that we refer to as population I, six populations are identified in a clockwise manner, whereby population VI (CD69-negative, CCR7$^{int}$) is marked in red (left). Contour plots are representative of 4 independent experiments. Bar graph shows the frequency of population VI cells of the indicated mice. Data are a

summary of 3 independent experiments. One-Way Anova $P < 0.0001$. Unpaired two-tailed Student's *t*-test was used to calculate *P* values. **, $P < 0.01$, and ***, $P < 0.001$. Data are presented as mean values ± SD. **D** Contour plots show the CD4 versus CD8 profiles of population V (top) and VI (bottom) from 2D2 and 2D2.CD103$^{Tg}$β7$^{Tg}$ mice. Results are representative of 3 independent experiments. **E** Histograms show surface CD103 (top) and nuclear Helios expression (bottom) of population V CD4 T cells and population VI DN cells of 2D2 thymocytes. Results are representative of 3 independent experiments. **F** Histograms show PD-1 expression on population V CD4 T cells and population VI DN cells of 2D2 and 2D2.CD103$^{Tg}$β7$^{Tg}$ thymocytes. Results are representative of 3 independent experiments. **G** Histograms show the intracellular staining for active caspase-3 in population V CD4 T cells and population VI DN cells of 2D2 thymocytes. Bar graphs are a summary of 3 independent experiments with 2D2 mice ($n = 5$). Unpaired two-tailed Student's *t*-test was used to calculate *P* values. ***, $P < 0.001$. Data are presented as mean values ± SD. **H** The frequencies of population VI cells were assessed among CXCR4-negative DN thymocytes of WT and CD103-deficient (*Itgae$^{-/-}$*) mice. Contour plots are representative and bar graph is a summary of 4 independent experiments with a total of 6 *Itgae$^{-/-}$* mice. Unpaired two-tailed Student's *t*-test was used to calculate *P* values. *, $P < 0.05$. Data are presented as mean values ± SD.

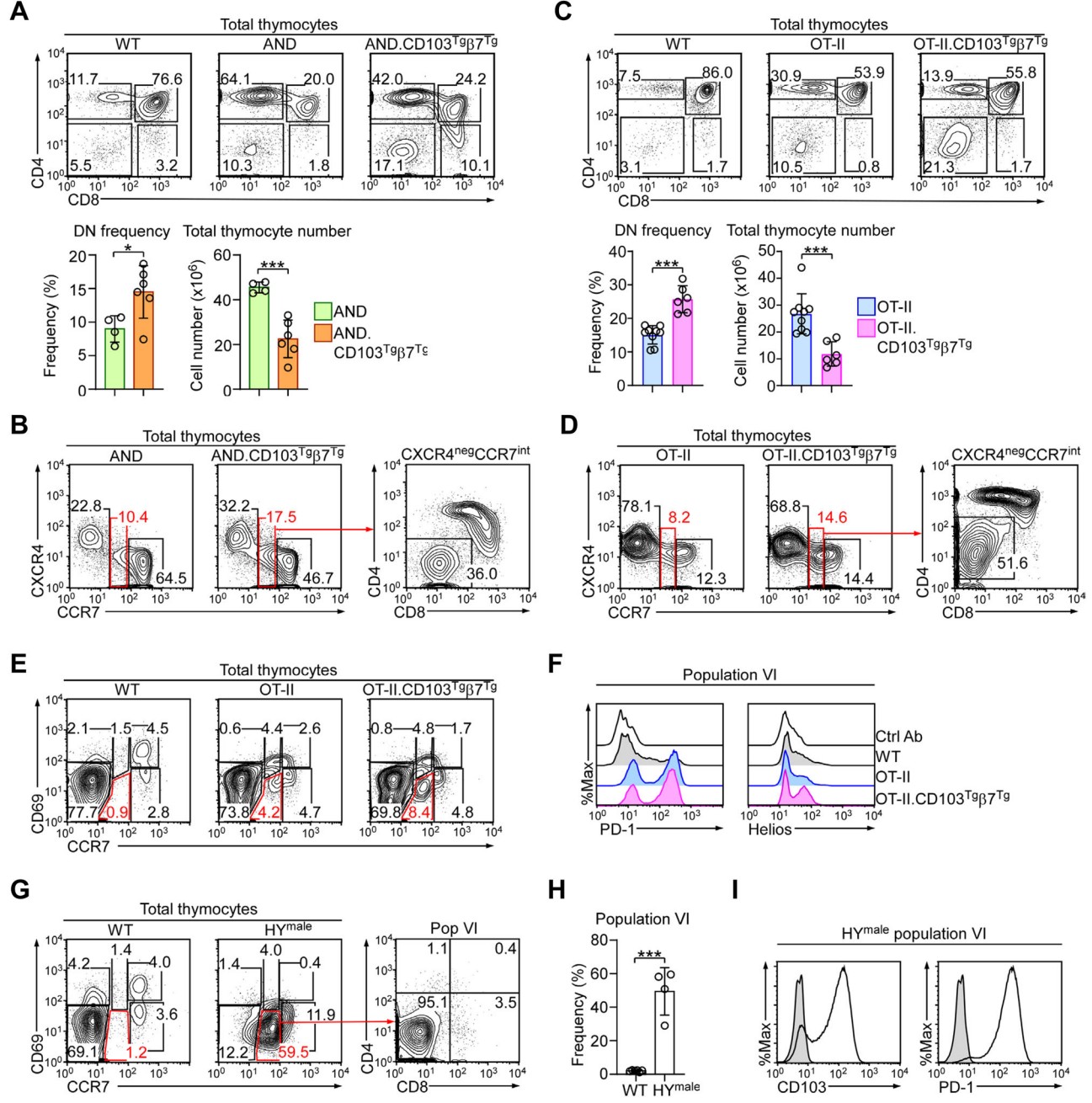

**Fig. 4 | Negative selection in OT-II and AND TCR transgenic mice. A** Contour plots (top) show CD4 versus CD8 profiles of WT, AND, and AND.CD103$^{Tg}$β7$^{Tg}$ thymocytes. Bar graphs (bottom) show the frequencies of DN and total thymocyte numbers of AND ($n = 4$) and AND.CD103$^{Tg}$β7$^{Tg}$ ($n = 6$) mice as summarized from 4 independent experiments. Unpaired two-tailed Student's *t*-test was used to calculate *P* values. *, *P* < 0.05 and ***, *P* < 0.001. Data are presented as mean values ± SD. **B** Contour plots show the CXCR4 versus CCR7 profiles of AND and AND.CD103$^{Tg}$β7$^{Tg}$ thymocytes (left), and the CD4 versus CD8 profile of CCR7$^{int}$ thymocytes of AND.CD103$^{Tg}$β7$^{Tg}$ mice (right). Results are representative of 5 independent experiments. **C** Contour plots (top) show CD4 versus CD8 profiles of WT, OT-II, and OT-II.CD103$^{Tg}$β7$^{Tg}$ thymocytes. Bar graphs (bottom) show the frequencies of DN and total thymocyte numbers of OT-II ($n = 9$) and OT-II.CD103$^{Tg}$β7$^{Tg}$ ($n = 6$) mice as summarized from 8 independent experiments. Unpaired two-tailed Student's *t*-test was used to calculate *P* values. ***, *P* < 0.001. Data are presented as mean values ± SD. **D** Contour plots show the CXCR4 versus CCR7 profiles of OT-II and OT-II.CD103$^{Tg}$β7$^{Tg}$ thymocytes (left), and the CD4 versus CD8 profile of CCR7$^{int}$ thymocytes of OT-II.CD103$^{Tg}$β7$^{Tg}$ mice (right). Results are representative of 6

independent experiments. **E** Contour plots show the CD69 versus CCR7 profiles of WT, OT-II, and OT-II.CD103$^{Tg}$β7$^{Tg}$ thymocytes, identifying six distinct populations whereby population VI is outlined in red. Results are representative of 3 independent experiments. **F** Histograms show surface PD-1 (left) and nuclear Helios expression (right) of population VI cells of WT, OT-II, and OT-II.CD103$^{Tg}$β7$^{Tg}$ thymocytes. Results are representative of 3 independent experiments. **G** Contour plots of total thymocytes (left) show the CD69 versus CCR7 profiles of WT and HY TCR transgenic male mice (HY$^{male}$), whereby population VI is outlined in red. The CD4 versus CD8 profile (right) was further determined on population VI cells of HY$^{male}$ mice. Results are representative of 2 independent experiments. **H** Bar graph shows the frequency of population VI cells among WT ($n = 6$) and HY$^{male}$ ($n = 4$) mice. Results are a summary of 2 independent experiments. Unpaired two-tailed Student's *t*-test was used to calculate *P* values. ***, *P* < 0.001. Data are presented as mean values ± SD. **I** Histograms show surface CD103 (left) and PD-1 expression (right) on population VI cells in HY$^{male}$ thymocytes. Results are representative of 2 independent experiments.

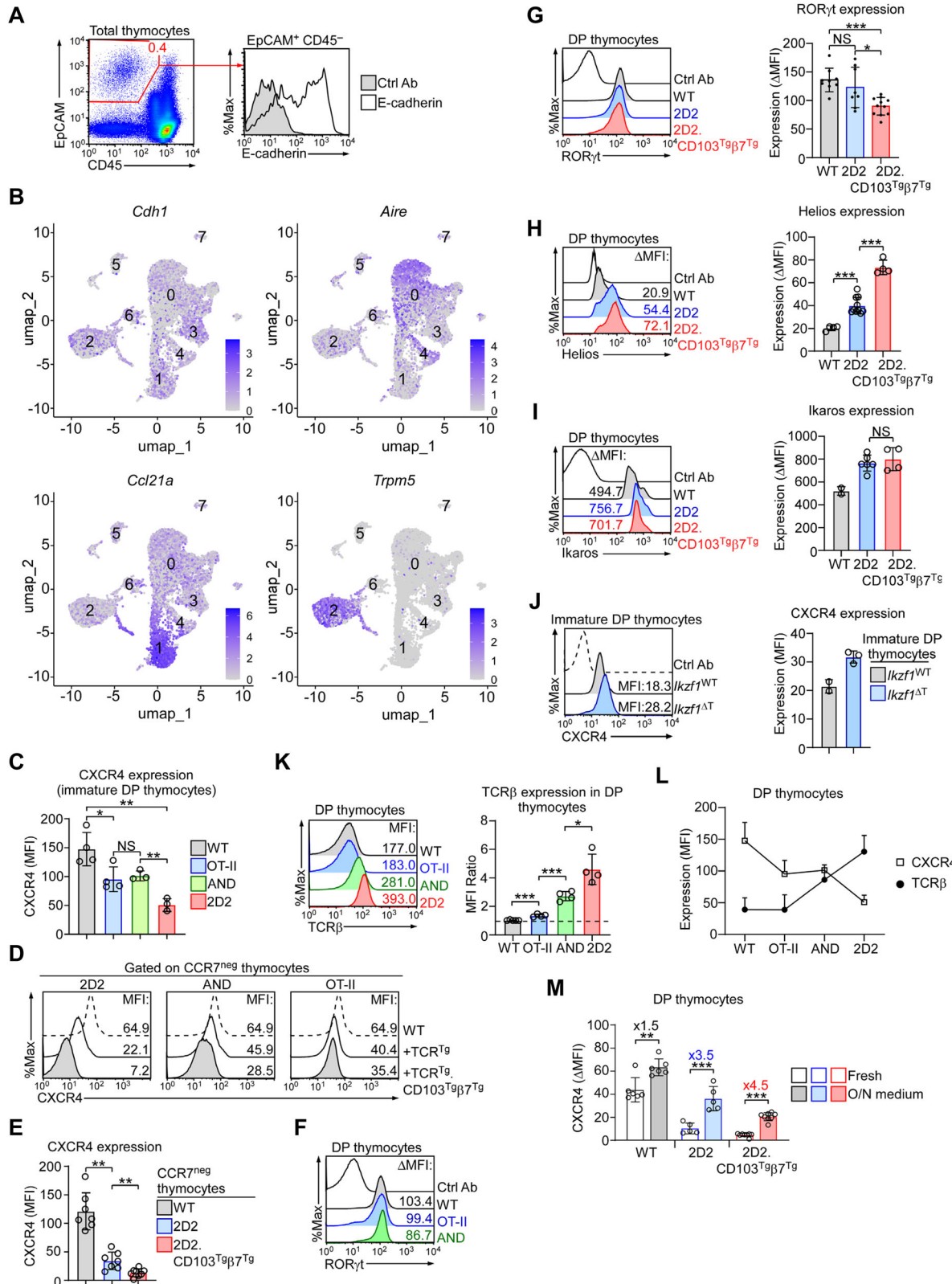

E-cadherin expression on mTECs is consequential because it would promote the binding of CD103+ thymocytes and facilitate the survey for autoreactive clones. On the other hand, pre-selection thymocytes are sequestered in the thymic cortex, such that the transgenic expression of CD103 alone would be insufficient to promote their engagement with mTECs. In this regard, we noticed that CXCR4 expression was significantly decreased in immature 2D2 thymocytes

(Fig. 3A). Because CXCR4 is critical to retain pre-selection thymocytes in the thymic cortex[29], the loss of CXCR4 would permit their migration into the medulla where E-cadherin-expressing mTEC are present (Fig. 5B)[37]. Consequently, immature 2D2 thymocytes could gain premature access to mTECs, and CD103 overexpression would promote their binding to and negative selection by mTECs. Along these lines, we noted that CXCR4 levels were significantly decreased on DP

**Fig. 5 | Transgenic TCR expression impacts CXCR4 chemokine receptor expression. A** E-cadherin expression on thymic epithelial cells that were isolated by protease digestion of thymus tissues and further identified by the absence of CD45 but high-level expression of EpCAM. Results are representative of 3 independent experiments. **B** UMAP plots of single-cell clusters generated by reanalyzing public mTEC scRNA-Seq data that identify 8 distinct populations. Relative expression of the indicated genes (*i.e.*, *Cdh1*, *Aire*, *Ccl21a*, and *Trpm5*) are indicated for each cluster. **C** Bar graph shows the protein abundance of CXCR4 (MFI) on CXCR4+CD24+CCR7−CD69− immature DP thymocytes of WT ($n = 4$), OT-II ($n = 4$), AND ($n = 3$), and 2D2 ($n = 3$) transgenic mice. Results are a summary of 3 independent experiments. Unpaired two-tailed Student's *t*-test was used to calculate *P* values. *, $P < 0.05$; **, $P < 0.01$; NS, not significant. One-Way Anova $P = 0.001$. Data are presented as mean values ± SD. **D** Histograms show the CXCR4 expression on CCR7neg pre-selection thymocytes of WT, 2D2, AND, and OT-II transgenic mice in the presence or absence of CD103Tgβ7Tg. Results are representative of 6 independent experiments. **E** Bar graph shows the protein abundance of CXCR4 (MFI) on CCR7neg thymocytes of WT ($n = 7$), 2D2 ($n = 7$), and 2D2.CD103Tgβ7Tg ($n = 8$) mice. Results are a summary of 5 independent experiments. Unpaired two-tailed Student's *t*-test was used to calculate *P* values. **, $P < 0.01$. One-Way Anova $P < 0.0001$. Data are presented as mean values ± SD. **F, G** Histograms show intracellular staining for RORγt in DP thymocytes of WT, OT-II, and AND transgenic mice (**F**) and of WT, 2D2, and 2D2.CD103Tgβ7Tg mice (**G**). Results are representative of 3 independent experiments with at least 3 mice per group. Ctrl Ab: control antibody. Bar graph shows the protein abundance of RORγt expression (ΔMFI) on DP thymocytes of WT ($n = 9$), 2D2 ($n = 8$), and 2D2.CD103Tgβ7Tg ($n = 11$) mice. Results are a summary of 5 independent experiments. Unpaired two-tailed Student's *t*-test was used to calculate *P* values. *, $P < 0.05$; ***, $P < 0.001$; NS, not significant. One-Way Anova

$P = 0.0026$. Data are presented as mean values ± SD. **H** Histogram (left) is representative and bar graph (right) shows the summary of Helios expression (ΔMFI) in DP thymocytes of WT ($n = 4$), 2D2 ($n = 10$), and 2D2.CD103Tgβ7Tg ($n = 4$) mice. Results are a summary of 3 independent experiments. Unpaired two-tailed Student's *t*-test was used to calculate *P* values. ***, $P < 0.001$. One-Way Anova $P < 0.0001$. Data are presented as mean values ± SD. **I** Histogram (left) is representative and bar graph (right) shows the summary of Ikaros expression (ΔMFI) in DP thymocytes of WT ($n = 2$), 2D2 ($n = 6$), and 2D2.CD103Tgβ7Tg ($n = 4$) mice. Results are a summary of 2 independent experiments. Unpaired two-tailed Student's *t*-test was used to calculate *P* values. **, $P < 0.01$; NS, not significant. One-Way Anova $P = 0.0084$. Data are presented as mean values ± SD. **J** Histogram (left) is representative and bar graph (right) shows the summary of CXCR4 expression (MFI) on immature DP thymocytes of *Ikfz1*WT and *Ikfz1*ΔT mice. Results are a summary of 2 independent experiments with 2 *Ikfz1*WT and 3 *Ikfz1*ΔT mice. Data are presented as mean values ± SD. **K, L** Histogram is representative (left) and bar graph (right) shows a summary of TCRβ expression (MFI ratio) on DP thymocytes of WT ($n = 5$), OT-II ($n = 4$), AND ($n = 4$), and 2D2 ($n = 4$) transgenic mice (**K**). The line graph shows the inverse correlation between TCRβ expression (MFI) and CXCR4 (MFI) in DP thymocytes of the indicated mice (**L**). Results are the summary of 4 independent experiments. Unpaired two-tailed Student's *t*-test was used to calculate *P* values. *, $P < 0.05$; ***, $P < 0.001$. One-Way Anova $P < 0.0001$. Data are presented as mean values ± SD. **M** Bar graph shows surface CXCR4 expression (ΔMFI) on freshly isolated or overnight (O/N) medium-cultured DP thymocytes of WT ($n = 6$), 2D2 ($n = 5$), and 2D2.CD103Tgβ7Tg ($n = 8$) mice. Results are the summary of 3 independent experiments. Paired two-tailed Student's *t*-test was used to calculate *P* values. **, $P < 0.01$; ***, $P < 0.001$. Data are presented as mean values ± SD.

---

thymocytes of other MHC-II restricted TCR transgenes as well (Fig. 5C), which was exacerbated by CD103 overexpression as assessed on pre-selection (*e.g.*, CCR7neg) thymocytes (Fig. 5D). Thus, the immature thymocytes of MHC-II restricted TCR-transgenic (TCRTg) mice could gain access to mTECs independently of positive selection. These results could explain how the premature expression of CD103 can promote the negative selection of TCRTg thymocytes, and specifically of 2D2 cells, which express the lowest amount of CXCR4 (Fig. 5D, E).

To understand why TCRTg thymocytes express low levels of CXCR4, we tested the possibility of a transcriptional mechanism controlling CXCR4 in TCRTg thymocytes. Specifically, we asked if the protein abundance of transcription factor RORγt would differ between WT and TCRTg DP thymocytes because we previously reported that RORγt is selectively induced on immature DP cells to control cytokine receptor expression[38]. Intranuclear staining for RORγt, however, revealed no differences in RORγt levels between WT and different TCR transgenes (Fig. 5F), and also not by overexpression of CD103 in 2D2 mice (Fig. 5G). Because strong TCR signaling induces Helios expression[32], we next considered the possibility of increased Helios expression as the molecular basis of reduced CXCR4 expression in 2D2 immature thymocytes. Interestingly, Helios expression was substantially increased in 2D2 DP thymocytes and further upregulated upon CD103 overexpression (Fig. 5H). Helios is a member of the Ikaros transcription factor family that cooperates with Ikaros in gene regulation[39], prompting us to examine the expression of Ikaros as well. Again, we found a significant increase in Ikaros expression in 2D2 DP thymocytes, supporting a previously unappreciated role for Ikaros family members in association with CXCR4 expression (Fig. 5I). To test whether Ikaros expression would only correlate or would be necessary for CXCR4 expression, we next generated mice with conditional deletion of Ikaros in all T lineage cells, *i.e.*, *Ikzf1*ΔT. Consistent with a potential negative regulatory role of Ikaros in CXCR4 expression, immature DP thymocytes of *Ikzf1*ΔT mice showed elevated levels of CXCR4 (Fig. 5J), affirming that Ikaros, and presumably Helios as well, is required to suppress CXCR4 expression. At this point, it remains unclear to us how transgenic TCRs would induce Helios/Ikaros expression to downregulate CXCR4. Nonetheless, we observed an inverse correlation of CXCR4 with TCRβ levels, whereby WT DP thymocytes expressed the largest amounts of CXCR4 but the

lowest levels of TCRβ (Fig. 5K, L). 2D2 DP thymocytes, on the other hand, expressed the lowest amounts of CXCR4 but the highest levels of TCRβ (Fig. 5K, L). Altogether, these results suggest that TCR engagement could regulate CXCR4 levels on DP cells through controlling Ikaros family transcription factors.

If such were the case, we postulated that disengaging the TCR would revert the suppression of CXCR4 expression on DP thymocytes. To this end, we isolated DP thymocytes from the thymus to disengage their TCRs and cultured them in vitro overnight in a single-cell suspension. Strikingly, removing the cells from the in vivo environment dramatically upregulated CXCR4 expression on both 2D2 and 2D2.CD103Tgβ7Tg cells (> 3.5-fold), and to a lesser degree on WT DP thymocytes (1.5 fold) (Fig. 5M). Collectively, these results document TCR expression and engagement as a regulatory mechanism of CXCR4 levels that potentially alter the intra-thymic trafficking and distribution of pre-selection thymocytes.

## Disorganized thymic architecture in 2D2 and 2D2.CD103Tgβ7Tg mice

To examine the effects of altered CXCR4 and TCR expression on intra-thymic trafficking, we next reviewed thymic sections of 2D2 mice by H&E histochemistry. Compared to that of WT mice, we found that the 2D2 thymic medulla was fragmented and disorganized in its morphology (Fig. 6A). Nonetheless, area calculation of the thymic cortex and medulla of 2D2 mice showed a quantitative increase in the cumulative medullary region that was concomitant to a decrease in the cortical region (Fig. 6B). Immunohistochemistry with mTEC- and cTEC-specific antibodies further confirmed the intact development of both mTEC and cTECs in these mice (Fig. 6C). Thus, 2D2 thymocytes are exposed to a relatively normal thymic environment despite the loss in CXCR4 and induced early TCR expression. The overexpression of CD103, however, had a striking effect on the thymic architecture of 2D2 mice with a blurred cortico-medullary junction (Fig. 6D, Suppl. Fig. 4C) as well as a more fragmented and less evident medullary area (Fig. 6D), resulting in a markedly decreased medulla to cortex ratio (Fig. 6D). In fact, mTECs appeared to be dispersed throughout the thymus (Suppl. Fig. 4D), suggesting that 2D2.CD103Tgβ7Tg thymocytes would gain easier access to mTECs for binding and thymic selection,

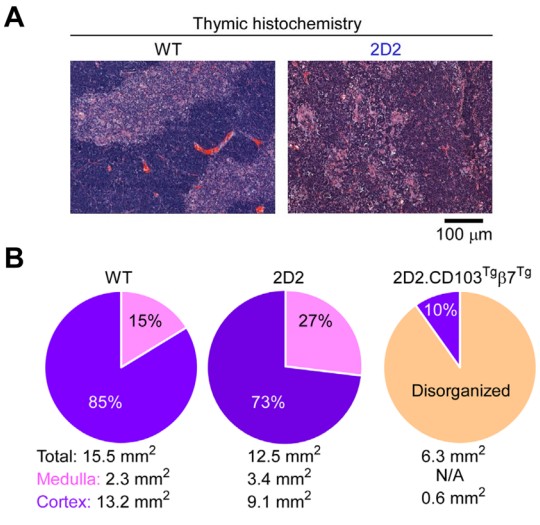

**A** Thymic histochemistry
WT    2D2
100 μm

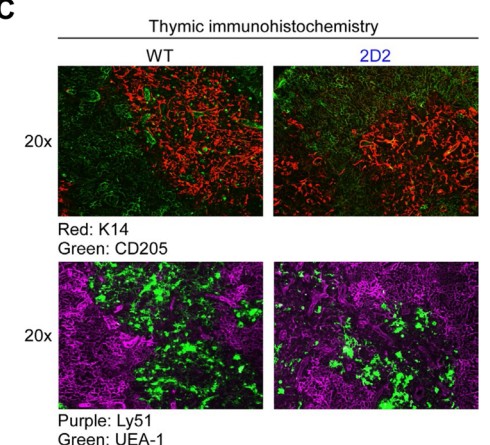

**B**
WT    2D2    2D2.CD103ᵀᵍβ7ᵀᵍ

15%    27%    10%
85%    73%    Disorganized

Total: 15.5 mm²    12.5 mm²    6.3 mm²
Medulla: 2.3 mm²    3.4 mm²    N/A
Cortex: 13.2 mm²    9.1 mm²    0.6 mm²

**C** Thymic immunohistochemistry
WT    2D2

20x

Red: K14
Green: CD205

20x

Purple: Ly51
Green: UEA-1

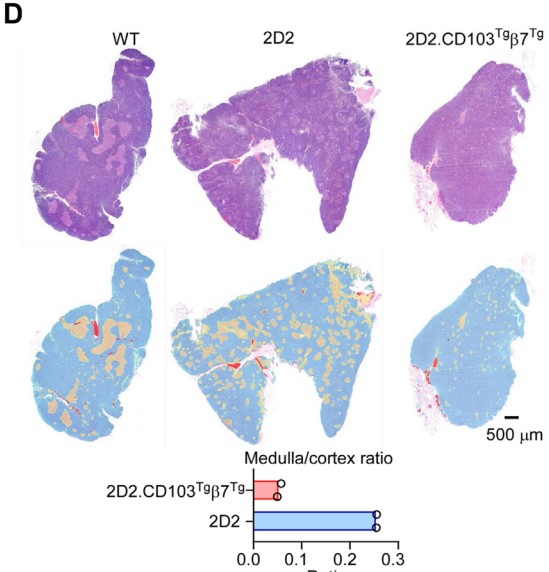

**D**
WT    2D2    2D2.CD103ᵀᵍβ7ᵀᵍ

500 μm

Medulla/cortex ratio
2D2.CD103ᵀᵍβ7ᵀᵍ
2D2
0.0   0.1   0.2   0.3
Ratio

**Fig. 6 | Characterization of the effect of CD103 on thymic architecture.**
**A** Histochemistry sections show representative hematoxylin and eosin (H&E)-stainings of the thymus of WT and 2D2 mice. Results are representative of two independent experiments. **B** Image scanning and medulla versus cortex area quantification of H&E-stained thymic sections of WT, 2D2, and 2D2.CD103ᵀᵍβ7ᵀᵍ mice using ImageJ Macro. Pie graphs shows the relative medullary and cortical areas for each mouse strains. The demarcation of the medullary area in 2D2.CD103ᵀᵍβ7ᵀᵍ mice was too blurry to be correctly identified by the ImageJ software such that it is considered as "disorganized". **C** Immunohistochemistry of WT and 2D2 thymic sections, stained with anti-K14 or -UAE-1 and anti-CD205 or -Ly51 to identify mTECs and cTECs, respectively. **D** Medulla and cortex area quantification of H&E-stained thymic sections of WT, 2D2, and 2D2.CD103ᵀᵍβ7ᵀᵍ mice by machine-learning random forest pixel classifier segmentation using Arivis Pro (v. 4.2) software. Whole thymic sections of the indicated mice were reimaged and reanalyzed using machine-learning software that permitted improved identification and segmentation of the cortical and medullary areas as shown in the bottom row of the image figure, cortex (cyan), medulla (yellow) and red blood cells (red). Bar graphs show the medulla/cortex ratio of the indicated mice, based on the scanned image areas. Data are presented as individual data points with the bar graphs showing the mean values.

of WT mice (Fig. 7A)[5]. CD103 overexpression in 2D2 mice, however, results in a significant decrease in peripheral T cells (Fig. 7A), which is likely to be associated with decreased thymic output of mature T cells in CD103 overexpressing 2D2 mice (Fig. 2C). Thus, compared to control 2D2 mice, the frequencies and numbers of total LN T cells in 2D2.CD103ᵀᵍ and 2D2.CD103ᵀᵍβ7ᵀᵍ mice are significantly reduced (Fig. 7A), which also indicates that peripheral homeostasis is impaired in these mice[40]. Interestingly, the CD103ᵀᵍ did not downregulate the frequency and number of positively selected 2D2 CD4 T cells (Fig. 7B) and all LN CD4 T cells also correctly expressed the clonotypic TCR (Suppl. Fig. 5A). Moreover, CD103 overexpression did not substantially affect the activation/differentiation status of 2D2 LN CD4 T cells (Fig. 7C). Accordingly, both 2D2.CD103ᵀᵍ and 2D2.CD103ᵀᵍβ7ᵀᵍ mice contained similar numbers of CD44ʰⁱCD62Lˡᵒ memory phenotype CD4 T cells (Fig. 7C), but with minimal effector functions based on their pro-inflammatory IFNγ expression (Suppl. Fig. 5B, C)[41,42]. On the other hand, the frequency and number of DN T cells, which would correspond to negatively selected T cells, were markedly decreased in 2D2.CD103ᵀᵍβ7ᵀᵍ mice (Fig. 7D). 2D2 DN T cells are autoreactive, as they produce copious amounts of IFNγ (Fig. 7E), contrasting to 2D2 CD4 T cells that are quiescent and express only minimal amounts of IFNγ (Suppl. Fig. 5B, C). Thus, the presence or the absence of the CD4 coreceptor identifies the products of positive versus negative selection, respectively.

Moreover, these results suggested that the thymic deletion of autoreactive DN T cells had lasting effects in peripheral tissues such that the autoreactive T cell compartment dramatically contracted upon CD103 overexpression (Fig. 7D). In fact, most of the remaining DN T cells in 2D2.CD103ᵀᵍβ7ᵀᵍ mice had lost their clonotypic TCRα Vα3.2 (Suppl. Fig. 5D), demonstrating the effective removal of autoreactive DN T cells from the peripheral T cell repertoire. Therefore, even as the frequency of IFNγ-producing DN T cells did not differ between 2D2 and 2D2.CD103ᵀᵍβ7ᵀᵍ mice (Fig. 7E), the absolute numbers of such pro-inflammatory autoreactive DN T cells markedly decreased upon CD103 overexpression. Thus, the forced expression of CD103 during thymic development has wide-ranging consequences that not only intensify the negative selection of autoreactive thymocytes, but also maintain self-tolerance in peripheral tissues.

### CD103 overexpression impairs the pro-inflammatory effect of 2D2 CD4 T cells
CD4 T cells normally lack CD103, which is also the case for 2D2 TCR transgenic CD4 T cells (Fig. 1I). To understand why CD103 is excluded from and whether its absence is necessary for CD4 T cells, we performed adoptive transfer experiments of in vitro activated 2D2 and

providing a potential explanation of enhanced negative selection by CD103 overexpression.

### CD103 overexpression is associated with a decrease in peripheral 2D2 DN T cells
2D2 thymocytes that escape negative selection populate peripheral lymphoid tissues to establish a T cell pool that is similar in size to that

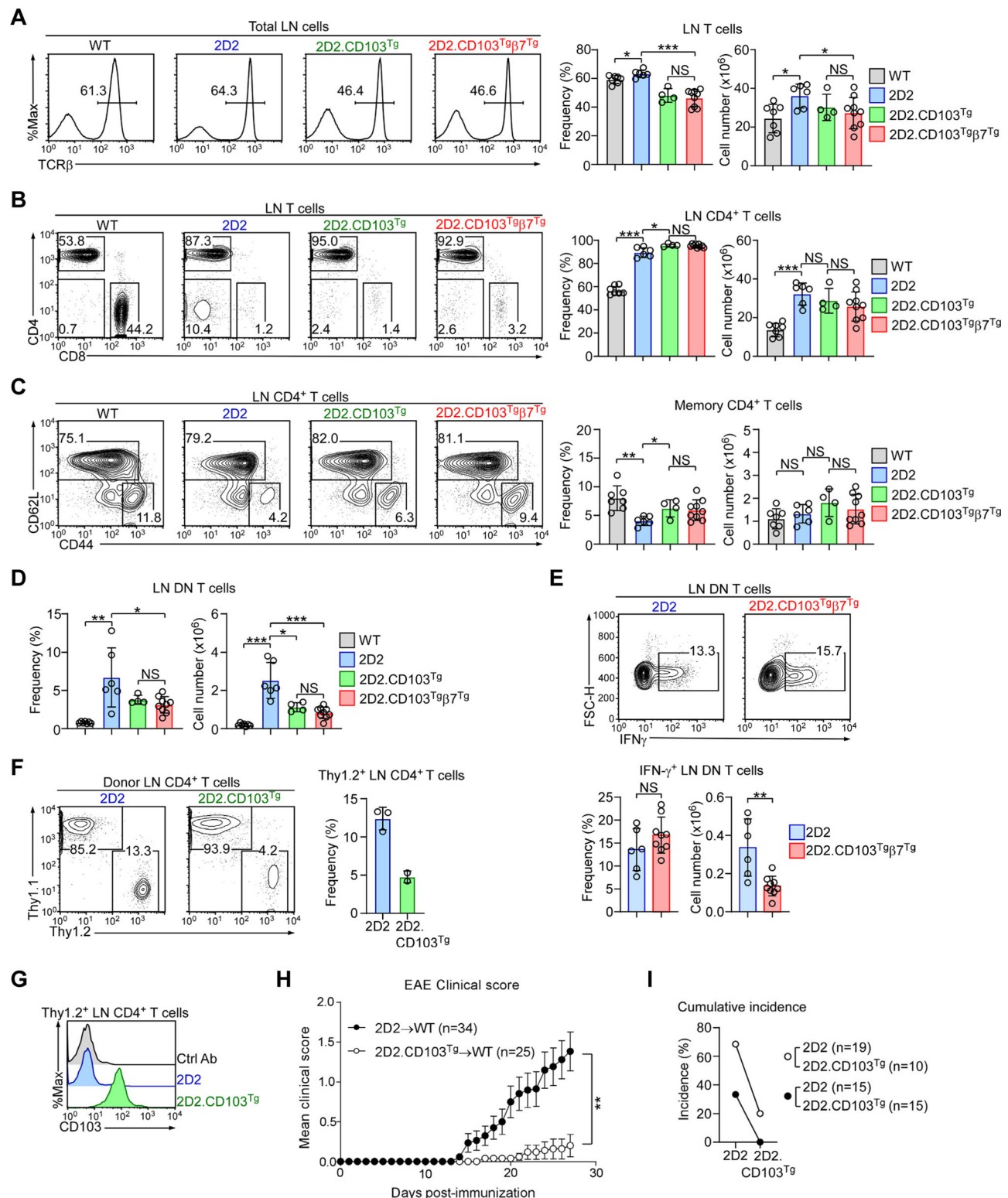

2D2.CD103$^{Tg}$ CD4 T cells into *Rag2*-deficient lymphopenic host mice. Importantly, we co-injected Thy1.1 congenic activated CD4 T cells at a 1:1 ratio with 2D2 or 2D2.CD103$^{Tg}$ CD4 T cells to normalize the adoptive transfer efficiency. Seven days after injection, donor T cells were recovered from host LNs and assessed for the relative frequencies of 2D2 versus 2D2.CD103$^{Tg}$ CD4 T cells (Fig. 7F). Interestingly, CD103 overexpression impaired the expansion of activated 2D2 cells, indicating that the ectopic expression of CD103, which we confirmed by

surface staining of donor T cells (Fig. 7G), is detrimental for the expansion and/or homeostasis of activated CD4 T cells. If this were the case, we postulated that we would observe the same impairment in a disease model. To this end, we employed the passive transfer model of EAE, in which purified 2D2 or 2D2.CD103$^{Tg}$ CD4 T cells were induced to differentiate into pathogenic Th17 cells in vitro and adoptively transferred into host mice for EAE disease induction[12]. As expected, 2D2.CD103$^{Tg}$ Th17 cells were markedly impaired in their ability to

**Fig. 7 | CD103 overexpression deletes autoreactive 2D2 DN T cells in the periphery. A** Histograms identify T cells in the LNs of WT ($n = 7$), 2D2 ($n = 6$), 2D2.CD103[Tg] ($n = 4$) and 2D2.CD103[Tg]β7[Tg] ($n = 9$) mice (left). Bar graphs show the frequency and number of LN T cells in the corresponding mice (right). Results are the summary of 3 independent experiments. Unpaired two-tailed Student's $t$-test was used to calculate $P$ values. *, $P < 0.05$; ***, $P < 0.001$; NS, not significant. Frequency One-Way Anova $P < 0.0001$; Cell number One-Way Anova $P = 0.0519$. Data are presented as mean values ± SD. **B** Contour plots show CD4 versus CD8 profiles of LN T cells of WT ($n = 7$), 2D2 ($n = 6$), 2D2.CD103[Tg] ($n = 4$) and 2D2.CD103[Tg]β7[Tg] ($n = 9$) mice (left). Bar graphs show the frequency and number of LN CD4⁺ T cells in the corresponding mice (right). Results are the summary of 4 independent experiments. Unpaired two-tailed Student's $t$-test was used to calculate $P$ values. *, $P < 0.05$; ***, $P < 0.001$; NS, not significant. Frequency One-Way Anova $P = 0.0003$; Cell number One-Way Anova $P < 0.0001$. Data are presented as mean values ± SD. **C** Contour plots show CD62L versus CD44 expression in LN CD4 T cells of WT ($n = 7$), 2D2 ($n = 6$), 2D2.CD103[Tg] ($n = 4$), and 2D2.CD103[Tg]β7[Tg] ($n = 9$) mice (left). Bar graphs (right) show the frequency and number of CD62L[lo] CD44[hi] memory phenotype CD4 T cells among LN CD4 T cells of the indicated mice. Results are the summary of 4 independent experiments. Unpaired two-tailed Student's $t$-test was used to calculate $P$ values. *, $P < 0.05$; **, $P < 0.01$; NS, not significant. Frequency One-Way Anova $P = 0.0042$; Cell number One-Way Anova $P = 0.1994$. Data are presented as mean values ± SD. **D** Bar graphs show the frequency and number of LN DN T cells in WT ($n = 7$), 2D2 ($n = 6$), 2D2.CD103[Tg] ($n = 4$), and 2D2.CD103[Tg]β7[Tg] ($n = 9$) mice (right). Results are the summary of 4 independent experiments. Unpaired two-tailed Student's $t$-test was used to calculate $P$ values. *, $P < 0.05$; **,

$P < 0.01$; ***, $P < 0.001$; NS, not significant. Frequency One-Way Anova $P < 0.0001$; Cell number One-Way Anova $P < 0.0001$. Data are presented as mean values ± SD. **E** Contour plots (top) show intracellular IFN-γ contents in PMA/ionomycin-stimulated LN DN T cells of 2D2 ($n = 6$) and 2D2.CD103[Tg]β7[Tg] ($n = 9$) mice. Bar graphs (bottom) show the frequency and absolute number of IFN-γ-expressing LN DN T cells in the indicated mice. Results show the summary of 4 independent experiments. Unpaired two-tailed Student's $t$-test was used to calculate $P$ values. **, $P < 0.01$; NS, not significant. Data are presented as mean values ± SD. **F** Contour plots show the recovery of in vitro activated donor CD4 T cells from *Rag2⁻/⁻* host mice, seven days after adoptive transfer (left). At the time of injection, donor T cells were comprised of 1:1 mixed WT (Thy1.1⁺) and 2D2 (Thy1.2⁺) or 1:1 mixed WT (Thy1.1⁺) and 2D2.CD103[Tg] (Thy1.2⁺) CD4 T cells. Bar graph shows the frequency of 2D2 or 2D2.CD103[Tg] CD4 T cells among total donor T cells (right). Results are the summary of a total of 5 adoptive transfer experiments of WT and 2D2 ($n = 3$) or WT and 2D2.CD103[Tg] ($n = 2$) mixed donor cells. Data are presented as mean values ± SD. **G** Histogram shows the CD103 expression on donor CD4 T cells of the indicated mice, seven days after adoptive transfer into *Rag2*-deficient host mice. **H** Clinical score of EAE-induced mice by adoptive transfer of in vitro activated 2D2 or 2D2.CD103[Tg] CD4 T cells. A total of 34 and 25 C57BL/6 mice were assessed for pathology upon injection of pathogenic 2D2 or 2D2.CD103[Tg] CD4 T cells, respectively. Two-Way ANOVA test was used to calculate the $P$ value. **, $P < 0.05$. Data are presented as mean values ± SD. **I** Graph shows the cumulative disease incidence that was assessed for each adoptive transfer group in 2 independent EAE experiments. Open circles are the results from the first EAE experiment, and closed circles show the results from the repeat experiment.

---

induce EAE compared to 2D2 Th17 cells (Fig. 7H), also resulting in substantially decreased disease incidence (Fig. 7I). Collectively, these results reveal a role of CD103 expression in pathogenic T cells, showing that the absence of CD103 is critical to preserve the effector function of mature CD4 T cells, while its expression reinforces the thymic negative selection process for immature thymocytes.

## Discussion

The elimination of an autoreactive TCR repertoire is achieved through the clonal deletion of T cells bearing such TCRs during T cell development in the thymus[3]. The 2D2 TCR is specific to the peptide 35-55 of the MOG extracellular domain, rendering 2D2 T cells to become autoreactive to self-MOG proteins in the context of MHC class II H-2[b] molecules[5]. Notably, 2D2 transgenic T cells somehow escape clonal deletion in the thymus, establishing a substantial population of mature 2D2 CD4 T cells both in the thymus and in peripheral tissues[5]. In fact, over 95% of peripheral T cells in 2D2 transgenic mice express this autoreactive TCR. However, only a small fraction (4%) of 2D2 mice develop spontaneous EAE and display symptoms of autoimmunity[7]. These findings support the notion that MOG antigens are inaccessible to 2D2 T cells under steady-state conditions, avoiding autoimmunity. It is presumably also the scarcity of MOG peptides in the thymus that could have resulted in incomplete clonal deletion of 2D2 T cells as there could be insufficient ligand density to achieve agonistic TCR signaling and negative selection[43]. Additionally, MOG 35-55 reactive CD4 T cells are cross-reactive to the neural antigen NF-M, such that 2D2 thymocytes could have escaped clonal deletion in the thymus due to the inefficient exposure to either of these self-antigens[44]. Altogether, a commonly accepted view on autoreactive 2D2 T cell generation proposes a mechanism that is based on insufficient availability of the negatively selecting antigen in the thymus.

In contrast, here, we provide evidence that the impaired clonal deletion in 2D2 TCR transgenic thymocytes could have been mediated by mechanisms independent of the abundance of thymic MOG peptides. Specifically, we show that negative selection can be reinforced by promoting their interaction with thymic epithelial cells which would alter the intra-thymic trafficking and thymic residency of 2D2 thymocytes. In our experimental model, thymic sequestration of 2D2 thymocytes was achieved through the thymocyte-specific overexpression of integrin CD103, which binds to E-cadherin that is expressed on

mTECs. Negative selection is usually mediated by a combination of thymic stromal cells and hematopoietic origin cells that include mTECs, dendritic cells, and B cells[45], among which mTECs play a prominent role[3]. In this regard, increasing the interaction with mTECs would facilitate the elimination of autoreactive 2D2 cells. Because such detrimental effects were specific to the autoreactive 2D2 TCR but less so for non-self-reactive AND and OT-II TCR transgenic thymocytes, our results also suggest that promoting the access to and engagement with mTECs can enhance negative selection.

The process of negative selection is anatomically located in the thymic medulla where tissue-restricted antigens (TRA) are presented to immature thymocytes[3,46,47]. TRAs are generated in mTECs by the transcriptional control of nuclear factors, such as Aire and Fezf2[48,49], and the presentation of these self-antigens is a critical process in educating immature thymocytes to discriminate between self and non-self[3]. Whether TRAs representing all tissues can be found in the medulla, and, if so, expressed in sufficient amounts to induce negative selection remains incompletely understood. But for MOG peptides, the literature suggests that MOG self-peptides are indeed available in the thymus[47], and that they are presented by mTECs and other thymic antigen-presenting cells. In this regard, MOG mRNA was found in purified mTECs but absent in thymic macrophages and dendritic cells[44]. Thymic B cells were also found to express MOG, indicating that MOG could be presented in the thymic microenvironment by various cells to mediate the deletion of MOG-specific T cells[50]. Consequently, MOG-specific T cells should have been removed from the peripheral repertoire, and the lack of MOG proteins, conversely, should permit the appearance of MOG-reactive T cells. To test this, MOG-deficient mice had been generated but were found to display no signs of neuronal defects or other abnormalities[43,51], despite the proposed role of MOG in myelination[8]. More importantly, testing T cell activation to MOG 35-55 peptides revealed markedly enhanced proliferation and Th1 cytokine production in MOG-deficient mice. These results indirectly suggested that MOG expression establishes tolerance to MOG in normal WT mice[51]. Nonetheless, it has been unclear how MOG-specific 2D2 T cells would escape clonal deletion in the presence of a functional negative selection mechanism. Our current data proposes a resolution to this conundrum by revealing that the 2D2 TCR transgene expression is associated with a dramatic decrease in CXCR4 expression, which is

typically highly expressed on preselection DP thymocytes[29]. Premature migration into the medulla due to the absence of CXCR4, however, would overwhelm the negative selection process by mTECs and other negative selecting hematopoietic origin cells in the medulla. If such were the case, we would postulate that many post-selection 2D2 thymocytes could exit the thymus without proper interaction with mTECs to scan their autoreactivity and commit them to negative selection. The forced expression of CD103, however, would promote thymocyte engagement with E-cadherin-expressing mTECs and restore effective negative selection, which is precisely what we observed in 2D2.CD103^{Tg}β7^{Tg} mice.

Altogether, the forced expression of CD103 has helped to uncover how 2D2 thymocytes could escape negative selection, which would be the result of changes in the intra-thymic trafficking and thymic retention of 2D2 T cells. Why 2D2 thymocytes would downregulate CXCR4 expression and perturb thymopoiesis remains to be investigated. Also, direct evidence for an increased binding of CD103-expressing 2D2 thymocytes and mTECs is not provided in our current study, and requires further assessments that we plan to address in our follow-up studies. Collectively, our results map the developmental pathway of autoreactive thymocytes using TCR transgenic mice, and propose that intra-thymic trafficking and distribution, followed by engagement with TRA-expressing mTECs could be a critical parameter in establishing central tolerance. Interestingly, AND and OT-II TCR transgenic thymocytes are less susceptible to CD103 overexpression, indicating that negative selection is contingent upon the TCR specificity, thereby affirming the antigen-specific nature of this thymic selection process.

## Methods

### Animals
C57BL/6 (B6) mice were obtained from the Charles River Laboratories (Frederick, MD). OT-II, 2D2, AND, *Rag2*[−/−], and *Itgae*[−/−] mice were procured from the Jackson Laboratories (JAX#004194, #006912, #002761, #008449 and #006144, respectively). T cell-specific CD103 transgenic (CD103^{Tg}) and integrin β7 (β7^{Tg}) transgenic mice were generated in-house and had been previously reported[17]. *Ikzf1*^{fl/fl} mice were described previously[52] and kindly provided by Drs. Chan and Kastner (INSERM, France), and were crossbred in-house with CD4^{Cre} mice procured from the Jackson Laboratories (#022071). HY TCR transgenic mice were described previously[34]. For all experiments, adult mice of both sexes, ranging from 6 to 20 weeks of age were used, with the exception for thymic epithelial cell characterization, where 2- to 3-week-old neonates were used. All mice were cared for in accordance with NIH guidelines under SPF conditions. Animals were housed in ventilated racks with an automatic watering system and *ad libitum* access to food on a 12-hour light/dark cycle under ambient conditions. At the end of the experiments, mice were euthanized by $CO_2$ inhalation. All animal procedures reported in this study that were performed by NCI-CCR-affiliated staff were approved by the NCI Animal Care and Use Committee (ACUC) and in accordance with federal regulatory requirements and standards. All components of the intramural NIH ACU program are accredited by AAALAC International.

### Flow cytometry
Fluorescence antibody-stained single-cell suspensions were analyzed using LSRFortessa or LSRII flow cytometers (BD Biosciences). For live cell analysis, dead cells were excluded by adding propidium iodide before running the samples on flow cytometers. For fixed cell staining and analysis, cells were stained with Ghost Dye Violet 510 (Cytek) for exclusion of dead cells, followed by surface staining and fixation with Foxp3 fixation buffer for transcription factors (eBioscience) or intracellular fixation buffer for cytokines (eBioscience). Afterwards, cells were permeabilized using the Foxp3 intracellular kit, following the manufacturer's instructions (eBioscience). Excess reagents were removed by extensive washing in FACS buffer (0.1% BSA, 0.1% sodium azide in HBSS) before flow cytometric analysis.

### Antibodies
Antibodies specific for the following antigens were used for staining: TCRβ (H57-597), CD4 (GK1.5), CD8α (53-6.7), CD8β (H35-17.2), CD25 (PC61.5), CD24 (M1/69), CD69 (H1.2F3), Vα3.2 (RR3-16), Vβ11 (RR3-15), CD103 (2E7), α4 (R1-2), β7 (FIB504), β1 (HMb1-1), CXCR4 (2B11), CCR7 (4B12), CD44 (IM7), CD62L (MEL-14), PD-1 (29 F.1A12), CD5 (53-7.3), MHC-I (AF6-88.5), IFNγ (XMG1.2), RORγt (Q31-378), CD45 (30-F11), E-Cadherin (DECMA-1), EpCAM (G8.8), gp38 (8.1.1), CD31(390), Helios (22F6), Ikaros (2A9), Ly6G/Ly6C (RB6-8C5), NK1.1 (PK136), γδ TCR (GL3), CD11b (M1/70), B220 (RA3-6B2), Active caspase-3 (FITC-DEVD-FMK), CD127 (A7R34), CD124 (mIL4R-M1), CD132(4G3), CD122 (TM-β1), and CD69 (H1.2F3), Thy1.1 (OX-7), Thy1.2 (53-2.1).

### Identification of DN thymocyte subsets
Single-cell suspension of total thymocytes was stained with a lineage antibody cocktail comprising biotinylated antibodies against CD8α (53-6.7), B220 (RA3-6B2), NK1.1 (PK136), CD11b (M1/70), Ly6G/Ly6C (RB6-8C5), and γδ TCR (GL3) for 40 minutes at 4 °C. Cells were washed in FACS buffer and then restained with fluorescence-conjugated anti-CD4 (GK1.5), anti-CD8β (H35-17.2), anti-CD25 (PC61.5), and anti-CD44 (IM7) antibodies for 40 minutes at 4 °C. Excess antibodies were washed out with FACS buffer, and cells were incubated with Alexa Fluor 594-conjugated streptavidin (Thermo) for 10 minutes at room temperature. Cells were then washed again in FACS buffer, and samples were acquired by flow cytometry. Dead cells were excluded by forward light scatter gating and propidium iodide staining. DN1-DN4 subsets were identified by CD25 versus CD44 expression among lineage-negative CD4, CD8 double-negative thymocytes as previously described[53].

### Isolation of thymic epithelial cells
Thymi were gently dissected into four uniform pieces and put into a 50 mL conical tube prelayered with 5 mL of the digestion solution (RPMI 1640 medium with 0.05% (w/v) of Liberase TH and 100 U/ml of DNase I). The tubes were then placed on a shaker at 37 °C for 20 minutes. After the complete digestion of the tissue, the supernatant was filtered into a new 14 ml conical tube and spun down to be resuspended in freshly prepared medium for cell counting. To identify thymic epithelial cells, the samples were stained with anti-CD45 and anti-EpCAM antibodies, whereby thymic epithelial cells are identified as CD45-negative EpCAM-positive cells.

### Immunofluorescence
Immunofluorescence was performed on tissue sections fixed with 4% paraformaldehyde (Cat # sc-281692, Santa Cruz Biotechnology) at 4 °C, followed by three changes of 10%, 20%, 30% sucrose solutions. Images were obtained by using a Nikon ECLIPS Ti2 (Nikon) and CSU-W1 confocal scanner unit (Yokogawa Electric Corporation) with a Plan-Apochromat 20×/0.8 objective and analyzed with NIS-Elements software (Nikon). Primary antibodies were directed against K14 (rabbit anti-mouse; Poly19053, Biolegend), CD205 (rat anti-mouse; 205yekta, Invitrogen), UEA-1-Biotin (Vector Laboratories), and Ly51-Alexa 647 (rat anti-mouse; 6C3, Biolegend). Secondary antibodies included goat anti-rat IgG Alexa 488, goat anti-rabbit IgG Alexa 568, and streptavidin-Alexa 488 (Invitrogen).

### In vitro culture of thymocytes
Freshly isolated thymocytes were resuspended in cell culture media at $2 \times 10^6$ cells/ml and plated into 24-well plates. Cells were cultured overnight either at 4 °C (fresh) or at 37 °C in a 7.5% $CO_2$ incubator before cell surface staining and analysis by flow cytometry.

### Adoptive transfer of in vitro activated CD4 effector T cells

Peripheral CD4 T cells from lymph nodes and spleen were purified from Thy1.1 mice and 2D2 or 2D2.CD103$^{Tg}$ mice using the MojoSort Mouse CD4 T Cell Isolation Kit (BioLegend) according to the manufacturer's protocol. The isolated CD4 T cells were plated in anti-CD3- and anti-CD28-coated plates and cultured in vitro in an incubator at 37 °C. The plated cells were resuspended in an IL-2-supplemented medium on day 3 and harvested on day 7 to be adoptively transferred to *Rag2*$^{-/-}$ host mice. The activated Thy1.1 and 2D2 or 2D2.CD103$^{Tg}$ CD4 T cells were mixed 1:1, resuspended in 200 μL PBS, and injected intravenously into the host mice. The adoptively transferred mice were analyzed on day 7 post-injection.

### Passive transfer model of experimental autoimmune encephalomyelitis

Naive CD4$^+$ T cells were isolated from 2D2 TCR transgenic mice (specific for MOG$_{35-55}$) and activated in vitro with soluble anti-CD3 antibodies (2 μg/ml; clone 145-2C11, BioXCell) in the presence of irradiated splenocytes (2,000 rad) at a 5:1 responder-to-APC ratio. Cells were cultured for 36 hrs in the presence of mouse IL-6 (30 ng/ml; Miltenyi Biotec), human TGF-β1 (3 ng/ml; Miltenyi Biotec), anti-IL-4 (10 μg/ml; clone 11B11, BioXCell), and anti-IFN-γ (10 μg/ml; clone XMG1.2, BioX-Cell) antibodies, followed by an additional 72 hrs in medium supplemented with recombinant mouse IL-23 (10 ng/ml; R&D Systems) to promote pathogenic T$_H$17 differentiation. On day 5, cells were re-stimulated for 48 hours on plates pre-coated with anti-CD3 (2 μg/ml; clone 145-2C11) and anti-CD28 (2 μg/ml; clone PV1, BioXCell) antibodies in the continued presence of IL-23 at 10 ng/ml. Fully differentiated T$_H$17 cells (7.5×10$^6$ cells per mouse) were adoptively transferred into C57BL/6 recipient mice via intravenous injection. EAE progression was monitored daily using a standard clinical scoring system: 0, no symptoms; 1, limp tail; 2, hind-limb weakness with impaired righting; 3, complete hind-limb paralysis; 4, forelimb and hind-limb paralysis; 5, moribund or deceased.

### Assessing intracellular cytokine production

For cytokine production analysis, freshly isolated lymph node cells were resuspended in cell culture media at 5 ×10$^6$ cells/ml density and stimulated with 50 ng/ml PMA (Sigma-Aldrich), 1 μM ionomycin (Sigma-Aldrich), and 3 μg/ml Brefeldin A (Invitrogen) for 3 hours. Cells were then washed once with FACS buffer followed by cell surface staining, fixation by IC Fixation Buffer (eBioscience), and permeabilization with Permeabilization Buffer (eBioscience), following the manufacturer's instructions. Fixed and permeabilized cells were then stained with intracellular antibodies for one hour, and analyzed with flow cytometry.

### Histology and light microscopy

The thymi were harvested and embedded in 2 ml of 4% paraformaldehyde (PFA) diluted in PBS, followed by incubation on a shaker at 4 °C overnight. The next day, tissue samples were shipped to Histoserv Inc (Gaithersburg, MD), where they were further processed for histochemistry preparation. In brief, PFA solution was removed and the fixed tissues were dehydrated through graded alcohols, cleared in xylene and then infiltrated with paraffin. After processing, the tissues were embedded in paraffin. The paraffin blocks were cut on a microtome at 5 μm. The unstained slides were deparaffinized through xylene and graded alcohols to water, stained in Carazzi's hematoxylin, then rinsed again in water. The slides were then placed in 95% ethanol before staining in eosin-phloxine and dehydrating through graded alcohols to xylene. The stained hematoxylin and eosin slides were then coverslipped using a permanent mounting medium. The sections were analyzed by microscopy and photographed using ZEN Microscopy Software (ZEISS).

### Histology image analysis by Arivis Pro

Brightfield tiled extended field of view images were collected using a Zeiss AxioObserver Z1 microscope equipped with a 10x plan-apochromat (N.A. 0.45) objective lens, AxioCam MRc5 CCD camera, X-Y motorized scanning stage and Zen software. The tiled images were stitched using Zen and then analyzed using Arivis Pro (v. 4.2.2; Carl Zeiss Microscopy, Thornwood, NY) image analysis software. Briefly, images were downsampled and a machine learning pixel classifier (random forest algorithm) was trained to segment four different components of the image; regions representative of thymic cortex, thymic medulla, red blood cells, and background extrathymic tissue. The trained algorithm was used in an analysis pipeline to segment the four components in the extended field of view images of the entire thymus sections, and the total area of each representative region was measured. The ratio of medulla to cortex area was then calculated. Separate pixel classifier models were trained for each of the sample genotypes.

### Histology image analysis by ImageJ Macro

Hematoxylin and eosin-stained thymus slices were imaged on Zeiss AxioObserverZ1 microscope equipped with 10x/0.45 Plan Apochromat objective, AxioCam MRc5 and mechanical XY scanning stage. Resulting tiles were combined into mosaic using custom ImageJ macro (www.github.com/janwisn/Histological_Mosaic_Assembly_and_Annotation), then area of cortex and medulla was measured after segmentation with another custom macro available at www.github.com/janwisn/Thymus_Segmentation. Segmentation thresholds for tissue and blood vessel detection as well as medulla/cortex differentiation (set as defaults in the above macro) were established based on brightness and/or color balance analysis of wild-type thymus image.

### Reanalysis of public single-cell RNA sequencing data

The single-cell RNA-Seq data used in this study is publicly available from the Gene Expression Omnibus (GEO) under accession number GSE241742[54]. The processed data from control mice were reanalyzed using Seurat (v5.1.0)[55] in R language (v4.4.2). Low-quality cells with a high percentage of mitochondrial genes were filtered out. Reads were log-normalized using the NormalizeData function, and cell cycle effects were regressed out using the ScaleData function, followed by the FindMultiModalNeighbors function, RunPCA function, and RunUMAP function (dims = 1:20) with default settings. The cells were clustered using FindClusters function, and the expression of feature genes for each cluster was visualized using the FeaturePlot function.

### Statistics

Data are shown as the mean values ± SD. Two-tailed Student's *t*-test was used to calculate *P* values. *P* values of less than 0.05 were considered significant, whereby *, $P < 0.05$; **, $P < 0.01$; and ***, $P < 0.001$; NS, not significant. One-way ANOVA test was used to calculate statistical significance when more than two groups were compared. The *P* values from the ANOVA tests are indicated in the respective figure legends. Statistical data were analyzed using the GraphPad Prism 8 software.

### Reporting summary

Further information on research design is available in the Nature Portfolio Reporting Summary linked to this article.

## Data availability

The data supporting the findings of this study are provided in the Figures, Supplementary Figs., or in the Source data file. All other data are available from the corresponding author upon request. Source data are provided with this paper.

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

## Acknowledgements

We thank Dr. Yousuke Takahama (NCI) for sharing his expertise on thymus biology and for providing critical feedback on our study. We thank Dr. Andy Tran (NCI) for his contributions in developing the AI model for segmenting the regions of the thymus. We also thank the members of the Park lab for their review of this manuscript. This study has been supported by the Intramural Research Program of the US National Institutes of Health, National Cancer Institute, Center for Cancer Research (ZIA BC 011214 and ZIA BC 011215).

## Author contributions

NL generated and analyzed data and contributed to the writing of the manuscript. C.L., M.A.L., H.R.K., J.L. and W.H. generated and analyzed data and reviewed the manuscript. J.W. and M.K. analyzed data and reviewed the manuscript. V.L. performed experiments, provided reagents, analyzed data and edited the manuscript. J.H.P. analyzed data, supervised the project and wrote the manuscript.

## Funding

## Competing interests

The authors declare no competing interests.
