## [Transparent Peer Review file · Nature Communications]

Integrin CD103 reveals a distinct developmental pathway of autoreactive thymocytes in TCR transgenic mice

Corresponding Author: Dr Jung Hyun Park

Version 0:

Reviewer comments:

Reviewer #1

(Remarks to the Author)

The study by Liman et al. attempts to investigate how autoreactive MHCII-restricted thymocytes evade negative selection. They propose that negative selection is associated with the integrin CD103 and show that the overexpression of CD103 suppresses CXCR4 expression and reinforces clonal deletion. In summary, although the authors suggest a new role of CD103 in negative selection, the relevance of their results, which are based on artificial systems, is questionable. In my opinion, this study is overall descriptive and suffers from a lack of clear demonstrations and mechanistic insights. At this stage the main results are not sufficiently convincing to represent a major breakthrough in the field and justify publication in Nature Communications.

Specific comments:

- As stated by the authors "How to evade negative selection, however, remains incompletely understood" is an important question. However, this study does not answer to this question, but rather shows that CD103 "reinforces clonal deletion".
- It is surprising that all the conclusions are based on CD103 overexpression and that the authors didn't use a conditional Ko of CD103 in thymocytes, to support their conclusions or at least compare their results of overexpression with those of CD103 ko mice.
- It would be important to define and show the impact of CD103 overexpression or deletion in a non-TCR transgenic system.
- It would also be important to use other autoreactive TCR transgenic mice to support their findings.
- The authors claim that "Thus, CD103 overexpression promotes negative selection in non-autoreactive TCR transgenes as well." And "Collectively, these results showed that CD103 overexpression profoundly alters the development of thymocytes, and they further suggested that the clonal deletion of autoreactive thymocytes can be reinforced by CD103-mediated mechanisms." How they reconcile the fact that CD103 overexpression promotes negative selection in mice that do not express the cognate antigen. These affirmations lack clear demonstrations of negative selection.
- How the authors reconcile the interpretation of their results with an absence of CD103 expression in DN, DP, CD4 SP thymocytes and a strong expression in only CD8 SP thymocytes in WT mice (see Fig. 2a) ?
- Mechanistically, the authors do not show how CD103 and CXCR4 are regulated.
- They speculate that the "importance of thymocyte interaction with E-cadherin-expressing thymic epithelial cells (TECs)" without showing the expression of E-cadherin in mTEC. Does a specific subset of mTEC express E-cadherin?
- They authors conclude that 2D2 TCR promotes the generation of mature TCRbhi CCR7+ MHC-I+ DN T cells, corresponding to negatively-selected cells without using reliable markers of negative selection such as cleaved caspase 3 (PMID: 31010850) or helios/PD1 markers (PMID: 23337809).
- Several conclusions such as "Examination of thymocyte maturation markers, i.e., CD5, CCR7, CD24, MHC-I, and CD69,

indicated that 2D2 CD4SP cells were fully differentiated and comparable to WT CD4 T cells (Suppl. Fig. 1A)” and “Thus, the generation of CD4 T cells in 2D2 mice appeared to be intact despite their autoreactivity, and comparable to that of non-self-specific CD4 T cells in AND mice” are over-interpreted. In many histograms, the MFI is not indicated and the expression is not quantified using several mice in several independent experiments. In Suppl. Fig. 1A, the authors should analyze the maturation status of 2D2 CD4+ SP thymocytes based on CD69/MHCI expression (PMID: 27043411).

- I strongly encourage the authors to show all the data indicated as “not shown” and in particular those related to the following sentence “we next assessed the expression of a series of developmental and differentiation markers by reanalyzing public RNAseq data sets of agonistic signaled thymocytes and screening custom-made panels of antibody arrays for surface molecules (data not shown)” justifying the analysis of CD103 in negative selection.

-The interpretation of results shown in Fig 4C should be reinforced by analyzing the localization of mTEC in these transgenic mice. It seems that the cortico-medullary demarcation is totally blurry in 2D2.CD103Tgb7Tg. The authors should quantify medullary areas in 2D2 and 2D2.CD103Tgb7Tg mice.

-The CD4/CD8 profile of 2D2 mice is not consistent in panels Fig. 1a and Fig.2d.

-Consider using an alternative approach to the t-tests that are used for some comparisons of more than two groups.

- statistics between WT and 2D2 are missing (Fig. 1B, C).

Reviewer #2

(Remarks to the Author)

Liman et. al. propose that CD103 expression is part of a developmental pathway for autoreactive thymocytes. Specifically, the authors propose that negative selection of MHCII-restricted thymocytes is associated with the induction of CD103 integrin. Further, overexpression of CD103 suppressed CXCR4 expression, which promoted clonal deletion. The authors studied 2D2 TCR transgenic mice, whereby T cells are specific to a MOG peptide, and are used as a model of neuritis/EAE caused by self-reactive T cell clones. In these mice, the currently accepted view is that self-reactive thymocytes escape negative selection because there is a low level of expression of the negative-selecting antigen in the thymus. In other words, developing thymocytes fail to encounter their negative-selecting antigen, are therefore not deleted, and leave the thymus as self-reactive T cells. The authors propose that instead, self-reactive 2D2 thymocytes undergoing agonist selection divert negative selection in a process that involves expression of CD103. This integrin promotes thymocyte interaction with E-cadherin expressed in medullary thymic epithelial cells. The authors show that the overall cortical/medullary areas are altered in 2D2 mice, and even more so in 2D2 overexpressing integrin CD103. The data is overall interesting, the models selected are informative and analyses are appropriate. There are, however, aspects in the manuscript that require additional attention. TCR transgenic mice can be useful to decipher specific aspects of T cell differentiation. Nevertheless, they often differ from wild type mice in the time at which they express their TCR, and that can affect the normal differentiation of those cells. It is important to clarify whether CD103 is relevant for negative selection in wild type mice, or to which extent it is a specificity of this transgenic TCR mouse line.

Specific comments:

Lines 119 to 122 and 137 to 139: Description of DN cells should include the normal DN stages using at least CD44 and CD25 to show when the Tg TCR starts to be expressed. That could help clarifying the stage from which differentiation might be altered. This can be important for data interpretation at the end of the same paragraph, where the authors state that mature DN bTCRhi result from negative selection. While the authors cite 2 papers to state that DN bTCRhi result from DP undergoing negative selection, the papers cited refer to CD8 α TCR $\alpha\beta$ precursors in the thymus, and the authors do not formally show that the same happens in the 2D2 thymocytes. I consider important to show experimentally that: 1) DN bTCRhi indeed result from DP and 2) bTCR expression in the normal DN stages of differentiation (DN1 to DN4).

Lines 192 to 195: interpretation lack formal demonstration that those thymocytes result from negative selection.

Lines 228 to 230: the statement “Thus, immature 2D2 thymocytes are equipped to engage mTECs because they reside in the correct anatomical location.” Requires formal demonstration. Immunohistological analyses with the appropriate markers are required to show what the authors state.

Lines 235 to 237: Sections depicted in Fig. 4C, Suppl. Fig. 4 are not clear and, by far, insufficient to support the author’s claims. Medullary areas should be shown and DN thymocytes that might be present in the medullas should be quantified. This is essential, especially if the authors want to keep their conclusion “that alters the intrathymic trafficking and distribution of preselection thymocytes which in turn can make them susceptible to increased negative selection upon CD103 overexpression” (in lines 262 to 264).

Minor comments:

Lines 102, 103: where the authors describe the similarities between the wt and the 2D2 thymocytes (Fig S1), also the differences should be stated.

Fig. 1H is not described in the text.

Lines 156 to 158: for clarity, please correct sentence to avoid “introduced” the transgenes. I would favor “crossed” the lines to generate...

Fig. 4A: specify “DP thymocytes” in the figure

Line 266: If cells do not exit the thymus, they don't get “depleted” in the periphery. This subtitle should be corrected for clarity.

Version 1:

Reviewer comments:

Reviewer #1

(Remarks to the Author)

I appreciate the comprehensive revisions made by the authors in response to my initial concerns about the descriptive nature of the study and the lack of mechanistic insights. The new data presented in this revised version substantially reinforce the conclusions of the manuscript. In particular, the use of other autoreactive TCR transgenic mice (HY and AND TCR transgenic mice) provides stronger support for the authors' conclusions. More importantly, the use of TCR non-transgenic CD103-deficient mice to examine negative selection provides more physiologically relevant evidence to support a role for CD103 in thymic selection. In this regard, the authors should highlight this point more clearly in the abstract. The new gating strategy based on CD69 and CCR7 is also more convincing in identifying negatively selected thymocytes.

Overall, these additions substantially improve the physiological relevance and novelty of the findings.

Minor points:

-I encourage the authors to show an H&E counterstaining in Figure 6A for the thymus of 2D2.CD103Tg.b7Tg mice at the same magnification as for the WT and 2D2 mice, given that the thymus of these mice is quantified in Figure 6B.

-Statistical analysis is missing in Figs 7H and 7I.

-Mean Fluorescence Intensity (MFI) values are not indicated in the histograms of several figures (Figs. 2A, 2B, 3E, 3F, 5D, and 5F).

Reviewer #2

(Remarks to the Author)

The authors considered and presented additional experiments that replied to all my comments and concerns. They indeed went well beyond the originally submitted data. I consider that the paper is very nice and recommend it to be accepted for publication.

#####

(Reviewer #2 was asked by the editor to mediate for reviewer #1, but later on reviewer #1 submitted their own report. After concurred by reviewer #2, the editor is posting below the mediating comment as a reference:)

#####

Mediating comments to author responses to R#1:

I consider that the substantial amount of new data provided in this revised version plus the corrections in the text fully address all concerns raised by Reviewer #1. I went through every single point raised, and consider that the revisions address every point. Finally, while the reviewer was indeed very critical of the manuscript, his/her remarks contributed to a better manuscript, that goes beyond the originally submitted version. I congratulate the authors for their work.

Below, you may find my point-by-point assessment, where my comment for every point raised is the 3rd, following the authors' reply.

The study by Liman et al. attempts to investigate how autoreactive MHCII-restricted thymocytes evade negative selection. They propose that negative selection is associated with the integrin CD103 and show that the overexpression of CD103 suppresses CXCR4 expression and reinforces clonal deletion. In summary, although the authors suggest a new role of CD103 in negative selection, the relevance of their results, which are based on artificial systems, is questionable.

We thank the reviewer for the succinct review and constructive criticisms. We greatly appreciate the comments, but we also respectfully disagree that the results from TCR transgenic mouse models would be physiologically irrelevant. In fact, the 2D2 TCR transgenic mice have spawned hundreds of publications (based on a PubMed search for “2D2” on April 4, 2025),

helping us to understand how autoimmunity is triggered and can be resolved. As such, we remain confident that employing genetically engineered mouse models to study fundamental mechanisms in immunology remains highly relevant.

This point is an opinion. The authors use a transgenic system to learn about the role of CD103 in negative selection. I consider that the point is well discussed in the paper and the limitations discussed.

In my opinion, this study is overall descriptive and suffers from a lack of clear demonstrations and mechanistic insights. At this stage the main results are not sufficiently convincing to represent a major breakthrough in the field and justify publication in Nature Communications.

We understand that the reviewer is concerned about the mechanistic insight and the broader impact of our findings. Prompted by the reviewer's comments, we now have performed further in-depth studies into the effect of CD103 in thymopoiesis using new mouse models and new experimental tools. Among others, we 1) examined the role of CD103 in negative selection using TCR non-transgenic CD103-deficient mice, 2) assessed E-cadherin expression on thymic epithelial cells (TECs) and mapped its distribution among medullary TECs (mTECs) by reanalyzing scRNA-Seq data, 3) initiated new collaborations to visualize and quantify changes in the thymic architecture by 2D2 and CD103 expression, and 4) performed adoptive transfer experiments and employed the Experimental Autoimmune Encephalomyelitis (EAE) model to demonstrate the physiological significance of CD103 expression or its absence in CD4 T cells.

Most importantly, we further refined the phenotype of DN thymocytes that have resulted from agonistic negative selection (Figure 1 for Reviewer 1).

In our original submission, we had used the CXCR4 versus CCR7 plots to identify negatively selected DN thymocytes (Figure 1 for Reviewer 1, top). During the revision, we found that assessing CD69 versus CCR7 expression of total thymocytes permits a much more precise identification of these cells that we found being highly enriched in population VI (Figure 1 for Reviewer 1, bottom). Consequently, we employed this new gating strategy to characterize the role of CD103 in the negative selection of 2D2 TCR transgenic mice in the revision, helping us to gain further insights into the mechanistic aspects of our findings.

I consider that the revised version addresses the concerns of the reviewer.

Specific comments:

-As stated by the authors "How to evade negative selection, however, remains incompletely understood" is an important question. However, this study does not answer to this question, but rather shows that CD103 "reinforces clonal deletion".

We agree that our question is inadequately formulated. We further appreciate the reviewer's comment that our study rather addresses a role for integrin CD103 in reinforcing thymic negative selection, such that this statement is misleading. We fixed this issue by rephrasing this statement and amended the revised manuscript accordingly (lines 39-41).

I consider that the revised version addresses the concerns of the reviewer.

-It is surprising that all the conclusions are based on CD103 overexpression and that the authors didn't use a conditional Ko of CD103 in thymocytes, to support their conclusions or at least compare their results of overexpression with those of CD103 ko mice.

To address the issue of not having tested the effects of CD103 deficiency, we obtained CD103-deficient mice and analyzed their thymocyte development. Of note, CD103 conditional KO mice had not been reported and are not available. However, the reviewer's comment is very much appreciated, and specific deletion of CD103 in immature thymocytes would be the ideal model to test the role of CD103 in thymic selection. Nonetheless, we consider generating conditional CD103-KO mice to be outside the scope of our current study, and we hope that the new results from CD103 germline KO mice will suffice to answer the reviewer's question (new Figure 3H). Specifically, we assessed CD103-KO thymocytes for the presence of population VI cells in the CD69 versus CCR7 plot, where we found them to be markedly reduced compared to WT thymocytes (new Figure 3H). These results are in full agreement with our model where CD103 reinforces negative selection in the thymus.

I consider that the revised version fully addresses the concerns of the reviewer.

-It would be important to define and show the impact of CD103 overexpression or deletion in a non-TCR transgenic system.

We agree with the reviewer, and we have now assessed the impact of CD103-deficiency in a non-TCR transgenic model. To this end, we obtained mice that are deficient in Itgae, the gene encoding CD103, and assessed the appearance of negatively selected DN thymocytes in comparison to those of WT mice. As mentioned above, we found that the CD69 versus CCR7 plot provides a higher resolution and more accurate identification of negatively selected DN thymocytes than our previous CXCR4 versus CCR7 plot (Figure 3A, new Figures 3B and 3C). Thus, we used the gating strategy based on CD69 and CCR7 to identify thymocytes undergoing negative selection and found a significant reduction of Population VI cells in CD103-deficient mice (new Figure 3H). These results agree with our model that CD103 expression reinforces negative selection in the thymus.

I consider that the revised version fully addresses the concerns of the reviewer.

- It would also be important to use other autoreactive TCR transgenic mice to support their findings.

This is an excellent point, as we would indeed expect to observe the same results in other autoreactive TCR models. To

this end, we acquired HY TCR transgenic mice whose TCR is specific to a male antigen [Kisielow P. et al., Nature, PMID: 3260350]. Consequently, HY TCR thymocytes are autoreactive in male mice. Strikingly, we found that the overwhelming majority of thymocytes in HY male mice resided within the population VI gate, which corresponds exactly to the region where we would expect to find negatively selected cells (new Figures 4G and 4H). Moreover, all of the population VI thymocytes in HY male mice were phenotypically DN (new Figure 4G), and they expressed high levels of CD103 and PD-1 (new Figure 4I). These results align with DN TCR β hi HY male thymocytes being derived from agonistic signaling-mediated negative selection.

To affirm this notion in another experimental model, we obtained AND TCR transgenic mice that were backcrossed from the H-2b background to the selecting I-Ek background and that express the agonistic PCC peptide 88-104 [Oehen S. et al., J Exp Med, PMID: 8676082] (kindly gifted by Dr. Xuguang Tai, CCR, NCI). In this thymic environment, the AND TCR becomes autoreactive. As shown below, the CD69 versus CCR7 plots of AND and AND I-Ek PCC+ mice were quite distinct, with a significant increase in negatively selected population VI cells of AND I-Ek PCC+ mice, concomitant with a decrease in population IV cells that are undergoing positive selection with increased TCR signaling intensity (Figure 2 for Reviewer 1). Collectively, these results from other autoreactive TCR transgenic mice further support our current findings.

I consider that the revised version fully addresses the concerns of the reviewer.

- The authors claim that "Thus, CD103 overexpression promotes negative selection in non-autoreactive TCR transgenes as well." And "Collectively, these results showed that CD103 overexpression profoundly alters the development of thymocytes, and they further suggested that the clonal deletion of autoreactive thymocytes can be reinforced by CD103-mediated mechanisms." How they reconcile the fact that CD103 overexpression promotes negative selection in mice that do not express the cognate antigen. These affirmations lack clear demonstrations of negative selection.

The reviewer raises a valid point that we wish to further elaborate on. Thymic positive selection is known to be mediated by weak TCR signaling, whereas negative selection is mediated by agonistic signaling. Also, positive selection is mediated by cTECs while negative selection is thought to occur in the medulla. Our new results on the thymic architecture of 2D2 TCR mice demonstrate that the 2D2 thymus becomes highly disorganized by the overexpression of CD103, resulting in immature thymocytes gaining premature access to mTECs (new Figures 6B, and 6D), enhancing their binding to E-cadherin expressing mTECs, which would promote TCR signaling to reinforce negative selection. In this regard, we have assessed the frequency and phenotype of OT-II and OT-II.CD103Tg β 7Tg population VI cells and found them to be substantially accumulated and contain a large fraction of PD-1 and Helios positive cells (new Figures 4E and 4F). Altogether, these results demonstrate that the forced expression of CD103 can mediate increased negative selection in support of our model. Demonstration of the effect on negative selection is difficult but I consider that the authors addressed the concerns of the reviewer.

-How the authors reconcile the interpretation of their results with an absence of CD103 expression in DN, DP, CD4 SP thymocytes and a strong expression in only CD8 SP thymocytes in WT mice (see Fig. 2a) ?

The selective expression of CD103 in post-selection CD8SP cells thymocytes is an intriguing observation that is controlled by the CD8-lineage specifying transcription factor, Runx3 [Grueter B. et al., J. Immunol., PMID: 16034110]. Our group has further expanded on this observation and recently reported a post-transcriptional mechanism of CD103 expression that requires the co-expression of and the heterodimerization with integrin β 7 for CD103 cell surface expression [Keller HR., et al., Cell Mol Life Sci, PMID: 34129058]. Moreover, we found that the limited availability of integrin β 7 in pre-selection immature thymocytes constrains CD103 expression during thymic development, such that CD103 expression is mostly limited to post-selection thymocytes. To clarify these points, we have expanded our manuscript on these issues (lines 170-172).

I consider the answer of the authors to be sufficient to address the comment.

- Mechanistically, the authors do not show how CD103 and CXCR4 are regulated.

The regulatory pathways of CD103 and CXCR4 expression have been previously mapped and were found to involve the transcription factors, Runx3 and E-proteins, respectively [Grueter B. et al., J. Immunol., PMID: 16034110; Kadakia T. et al., J Exp Med, PMID: 31201207]. Whether 2D2 and other TCR transgenes would employ the same mechanisms to induce CD103 upon agonistic selection and to downregulate CXCR4 on DP thymocytes, however, remains currently unclear to us. In this regard, the reviewer is correct that we did not further explore the molecular mechanisms of CD103 and CXCR4 expression in our current study.

Nonetheless, we are happy to share that we made some promising observations that are now added to the revision. Prompted by our previous observations that the transcription factor ROR γ t is highly and specifically expressed in immature DP thymocytes and that ROR γ t controls cytokine receptor expression in these cells [Ligons DL. et al., Science Signaling, PMID: 30154103], we assessed ROR γ t levels in DP thymocytes of different TCR transgenic mice to test a causal relationship with CXCR4 expression (Figure 5F). Unfortunately, we failed to observe any significant correlations between surface CXCR4 levels and ROR γ t expression, although we noticed a slight drop in ROR γ t levels upon the forced expression of CD103 (revised Figure 5G). Altogether, these results suggested that ROR γ t is unlikely to control CXCR4 expression downstream of transgenic TCRs. However, we found that the expression of Ikaros, which is another transcription factor that is highly upregulated in DP thymocytes [Mitchell JL. et al., Mol Immunol, PMID: 28376432], was significantly induced in 2D2 TCR transgenic DP cells (new Figure 5I). More importantly, the genetic deletion of Ikaros in T cells resulted in increased CXCR4 expression in DP thymocytes (new Figure 5J), suggesting that Ikaros could be acting as a transcriptional suppressor of CXCR4 expression. Thus, we consider it likely that 2D2 and other TCR transgenes could induce Ikaros

expression on pre-selection DP thymocytes to downregulate CXCR4 expression. Whether this is indeed the regulatory mechanism of CXCR4 downregulation in 2D2 thymocytes requires further analyses, which we aim to address in our follow-up studies.

The revised version fully addresses the concerns of the reviewer.

- They speculate that the “importance of thymocyte interaction with E-cadherin-expressing thymic epithelial cells (TECs)” without showing the expression of E-cadherin in mTEC. Does a specific subset of mTEC express E-cadherin?

These are two important and interesting issues that we now have addressed in the revision. First, we identified thymic epithelial cells in enzymatically digested thymic tissues using anti-EpCAM and anti-CD45 antibodies and assessed their E-cadherin expression. As shown in the new Figure 5A, E-cadherin was highly but also broadly expressed among TECs, indicating heterogeneity in E-cadherin expression among various TEC subsets.

Thus, we next aimed to map E-cadherin expression among different subsets of mTECs as requested by the reviewer. To this end, we reanalyzed public scRNA-Seq data of thymic epithelial cells for the expression of *Cdh1*, the gene encoding E-cadherin, in association with distinct markers of mTEC subsets, such as *Aire*, *Ccl21a*, but also *Trpm5*, which is associated with thymic Tuft cells (new Figure 5B). UMAP analyses identified 8 distinct populations, among which *Cdh1*-expressing cells were present in most clusters but specifically enriched in Tufts cells (cluster 2) and *Ccl21a*-expressing cells. Therefore, in agreement with the flow cytometry results, E-cadherin is evidently expressed in mTECs (new Figure 5A), but its abundance differs depending on the subset. We plan to follow up on the biological significance of these findings in our future studies.

The revised version fully addresses the concerns of the reviewer.

-They authors conclude that 2D2 TCR promotes the generation of mature TCR β hi CCR7+ MHC-I+ DN T cells, corresponding to negatively-selected cells without using reliable markers of negative selection such as cleaved caspase 3 (PMID: 31010850) or helios/PD1 markers (PMID: 23337809).

We agree that established markers of negative selection would be necessary to formally acknowledge TCR β hiCCR7+ DN thymocytes as negatively selected cells in 2D2 mice. To this end, we assessed surface PD-1, intracellular active caspase-3, and nuclear Helios expression as requested by the reviewer. Of note, in the revision, we were able to further refine the gating of negatively selected thymocytes by adding the surface marker CD69 for their identification, such that TCR β hi CCR7+ DN cells (in the original manuscript) are now TCR β hiCCR7^{int}CD69⁻ DN cells (also called population VI, henceforth). Positively selected CD4SP thymocytes, on the other hand, are identified as TCR β hiCCR7+CD69⁻ CD4SP cells (also called population V, henceforth). The modified gating strategies are shown in the new Figures 3B, 3C, and 3D, and the results for PD-1, active caspase-3, and Helios expression are shown in the new Figures 3F, 3G, and 3E, respectively. As expected, all markers associated with negative selection were highly induced in the negatively selected population VI but absent from positively selected population V cells, affirming that TCR β hiCCR7^{int}CD69⁻ DN thymocytes are the products of negative selection.

The revised version fully addresses the concerns of the reviewer.

-Several conclusions such as “Examination of thymocyte maturation markers, i.e., CD5, CCR7, CD24, MHC-I, and CD69, indicated that 2D2 CD4SP cells were fully differentiated and comparable to WT CD4 T cells (Suppl. Fig. 1A)” and “Thus, the generation of CD4 T cells in 2D2 mice appeared to be intact despite their autoreactivity, and comparable to that of non-self-specific CD4 T cells in AND mice” are over-interpreted.

We thank the reviewer for pointing out these issues. We have carefully re-evaluated our manuscript and rephrased these conclusions to avoid overinterpretations of the results. The changes are made in lines 109-112 where we changed the original sentence to “...surface markers that are associated with thymocyte selection and maturation, i.e., CD5, CCR7, CD24, MHC-I, and CD69, indicated that 2D2 CD4SP cells were comparable to WT CD4 T cells in their differentiation status.”. We also changed the conclusion in lines 120-122 to “Thus, the generation of CD4 T cells in 2D2 TCR transgenic mice appeared to be intact despite the autoreactivity of their TCR.”

The revised version addresses the concerns of the reviewer.

In many histograms, the MFI is not indicated and the expression is not quantified using several mice in several independent experiments.

In response to the reviewer’s request, we added the MFI to the histograms and quantified the expression. Specifically, we added the MFIs to the following figures; revised Figures 1J and 1I, new Figures 5H, 5I, and 5J, and revised Figure 5K, and the results were quantified in the associated bar graphs.

Addressed

In Suppl. Fig. 1A, the authors should analyze the maturation status of 2D2 CD4+ SP thymocytes based on CD69/MHC-I expression (PMID: 27043411).

This is an excellent point. As suggested, we employed the distinct expression of CD69 versus MHC-I among

CCR7+TCRβ+ post-selection thymocytes to analyze the maturation status of 2D2 CD4SP thymocytes. The results are added as new Suppl. Figure 3A, and they demonstrate that 2D2 CD4SP cells are mostly comprised of fully mature T cells.

Addressed

- I strongly encourage the authors to show all the data indicated as “not shown” and in particular those related to the following sentence “we next assessed the expression of a series of developmental and differentiation markers by reanalyzing public RNAseq data sets of agonistic signaled thymocytes and screening custom-made panels of antibody arrays for surface molecules (data not shown)” justifying the analysis of CD103 in negative selection.

As suggested, we removed all “data not shown” from the revised manuscript, such that now we only explain and discuss results displayed in the manuscript. In this regard, we added the results from screening surface molecules that are induced on negatively selected TCRβhi DN but not on positively selected TCRβhi CD4SP cells in 2D2 mice to the revision (new Suppl Figures 1D and 1E). These molecules include a series of cytokine receptors, activation markers, and integrins, and it is through this screening process that we identified the marked upregulation of integrin CD103 in TCRβhi DN thymocytes (revised Figure 1I).

Addressed

-The interpretation of results shown in Fig 4C should be reinforced by analyzing the localization of mTEC in these transgenic mice. It seems that the cortico-medullary demarcation is totally blurry in 2D2.CD103Tgβ7Tg. The authors should quantify medullary areas in 2D2 and 2D2.CD103Tgβ7Tg mice.

The reviewer is correct to point out the blurred cortico-medullary junction (CMJ) in 2D2 mice, which we found to be further exacerbated in 2D2.CD103Tgβ7Tg mice. By retaking the H&E histochemistry images (revised Figure 6A), we affirm that the blurriness is not due to low resolution of the images but due to fragmentation of the medulla and dispersion of the CMJ. For an independent evaluation of our findings, we initiated a new collaboration with Dr. Yousuke Takahama (National Cancer Institute, NIH), who helped us to map the mTEC and cTEC locations by immunohistochemistry, affirming the altered thymic architecture in 2D2 TCR transgenic mice (new Figure 6C). Also, as requested, we quantified the medullary areas of the 2D2 and 2D2.CD103Tgβ7Tg mice, for which we employed the help of Dr. Michael Kruhlak (CCR Confocal Microscopy facility, NCI, NIH). We quantified the medullary and cortical areas using Arivis Pro software (Zeiss) and then applied machine learning pixel classifiers (random forest algorithm) to determine the ratio of medulla to cortex. The results revealed a dramatically reduced medulla to cortex ratio in 2D2.CD103Tgβ7Tg mice, demonstrating the detrimental effect of CD103 overexpression in thymic development (new Figure 6D and Materials and Methods).

Addressed

-The CD4/CD8 profile of 2D2 mice is not consistent in panels Fig. 1a and Fig.2d.

We acknowledge that the CD4 versus CD8 profiles of 2D2 mice were inconsistent, and we have replaced them with correctly representative data, as shown in revised Figure 1A and Figure 2D.

Addressed

-Consider using an alternative approach to the t-tests that are used for some comparisons of more than two groups

We appreciate the suggestion. Following the reviewer’s suggestion, we tested the statistical power using one-way ANOVA for the Figures 1B, 1C, 1G, Figures 2C, 2D, 2E, Figures 3A, 3C, and Figures 5C, 5E, 5G, 5H, 5I, 5K and Figures 7A, 7B, 7C, and 7D, in the revision

Addressed

- statistics between WT and 2D2 are missing (Fig. 1B, C).

As requested, we added the statistics of cell numbers and frequencies between WT and 2D2 mice to the bar graphs in the revised Figures 1B and 1C.

Addressed

Reviewer #1 (Remarks to the Author)

The study by Liman et al. attempts to investigate how autoreactive MHCII-restricted thymocytes evade negative selection. They propose that negative selection is associated with the integrin CD103 and show that the overexpression of CD103 suppresses CXCR4 expression and reinforces clonal deletion. In summary, although the authors suggest a new role of CD103 in negative selection, the relevance of their results, which are based on artificial systems, is questionable.

→ We thank the reviewer for the succinct review and constructive criticisms. We greatly appreciate the comments, but we also respectfully disagree that the results from TCR transgenic mouse models would be physiologically irrelevant. In fact, the 2D2 TCR transgenic mice have spawned hundreds of publications (based on a PubMed search for “2D2” on April 4, 2025), helping us to understand how autoimmunity is triggered and can be resolved. As such, we remain confident that employing genetically engineered mouse models to study fundamental mechanisms in immunology remains highly relevant.

In my opinion, this study is overall descriptive and suffers from a lack of clear demonstrations and mechanistic insights. At this stage the main results are not sufficiently convincing to represent a major breakthrough in the field and justify publication in Nature Communications.

→ We understand that the reviewer is concerned about the mechanistic insight and the broader impact of our findings. Prompted by the reviewer’s comments, we now have performed further in-depth studies into the effect of CD103 in thymopoiesis using new mouse models and new experimental tools. Among others, we 1) examined the role of CD103 in negative selection using TCR non-transgenic CD103-deficient mice, 2) assessed E-cadherin expression on thymic epithelial cells (TECs) and mapped its distribution among medullary TECs (mTECs) by reanalyzing scRNA-Seq data, 3) initiated new collaborations to visualize and quantify changes in the thymic architecture by 2D2 and CD103 expression, and 4) performed adoptive transfer experiments and employed the Experimental Autoimmune Encephalomyelitis (EAE) model to demonstrate the physiological significance of CD103 expression or its absence in CD4 T cells.

Most importantly, we further refined the phenotype of DN thymocytes that have resulted from agonistic negative selection (**Figure 1 for Reviewer 1**).

Figure 1 for Reviewer 1

Top. Total thymocytes of WT and 2D2 TCR transgenic mice were assessed for expression of the chemokine receptors CXCR4 and CCR7. Based on the protein abundance of CCR7, we identified three distinct populations, among which the CCR7^{int} population (marked in red) contains negatively selected thymocytes.

Bottom. Alternatively, total thymocytes of WT and 2D2 mice were plotted for CD69 versus CCR7 expression, revealing 6 distinct populations of different developmental stages. Stages 1-5 had been previously described [PMID: 31201207]. In the current study, we identified a new population (pop. VI; marked in red) that mostly corresponds to negatively selected cells that are substantially enriched in 2D2 TCR transgenic thymocytes.

In our original submission, we had used the CXCR4 versus CCR7 plots to identify negatively selected DN thymocytes (**Figure 1 for Reviewer 1, top**). During the revision, we found that assessing CD69 versus CCR7 expression of total thymocytes permits a much more precise identification of these cells that we found being highly enriched in population VI (**Figure 1 for Reviewer 1, bottom**). Consequently, we employed this new gating strategy to characterize the role of CD103 in the negative selection of 2D2 TCR transgenic mice in the revision, helping us to gain further insights into the mechanistic aspects of our findings.

Specific comments:

-As stated by the authors “How to evade negative selection, however, remains incompletely understood” is an important question. However, this study does not answer to this question, but rather shows that CD103 “reinforces clonal deletion”.

→ We agree that our question is inadequately formulated. We further appreciate the reviewer’s comment that our study rather addresses a role for integrin CD103 in reinforcing thymic negative selection, such that this statement is misleading. We fixed this issue by rephrasing this statement and amended the revised manuscript accordingly (lines 39-41).

-It is surprising that all the conclusions are based on CD103 overexpression and that the authors didn't use a conditional Ko of CD103 in thymocytes, to support their conclusions or at least compare their results of overexpression with those of CD103 ko mice.

→ To address the issue of not having tested the effects of CD103 deficiency, we obtained CD103-deficient mice and analyzed their thymocyte development. Of note, CD103 conditional KO mice had not been reported and are not available. However, the reviewer’s comment is very much appreciated, and specific deletion of CD103 in immature thymocytes would be the ideal model to test the role of CD103 in thymic selection. Nonetheless, we consider generating conditional CD103-KO mice to be outside the scope of our current study, and we hope that the new results from CD103 germline KO mice will suffice to answer the reviewer’s question (**new Figure 3H**). Specifically, we assessed CD103-KO thymocytes for the presence of population VI cells in the CD69 versus CCR7 plot, where we found them to be markedly reduced compared to WT thymocytes (**new Figure 3H**). These results are in full agreement with our model where CD103 reinforces negative selection in the thymus.

-It would be important to define and show the impact of CD103 overexpression or deletion in a non-TCR transgenic system.

→ We agree with the reviewer, and we have now assessed the impact of CD103-deficiency in a non-TCR transgenic model. To this end, we obtained mice that are deficient in *Itgae*, the gene encoding CD103, and assessed the appearance of negatively selected DN thymocytes in comparison to those of WT mice. As mentioned above, we found that the CD69 versus CCR7 plot provides a higher resolution and more accurate identification of negatively selected DN thymocytes than our previous CXCR4 versus CCR7 plot (**Figure 3A, new Figures 3B and 3C**). Thus, we used the gating strategy based on CD69 and CCR7 to identify thymocytes undergoing negative selection and found a significant reduction of Population VI cells in CD103-deficient mice (**new Figure 3H**). These results agree with our model that CD103 expression reinforces negative selection in the thymus.

- It would also be important to use other autoreactive TCR transgenic mice to support their findings.

→ This is an excellent point, as we would indeed expect to observe the same results in other autoreactive TCR models. To this end, we acquired HY TCR transgenic mice whose TCR is specific to a male antigen [Kisielow P. *et al.*, Nature, PMID: 3260350]. Consequently, HY TCR thymocytes are autoreactive in male mice. Strikingly, we found that the overwhelming majority of thymocytes in HY male mice resided within the population VI gate, which corresponds exactly to the region where we would expect to find negatively selected cells (**new Figures 4G and 4H**). Moreover, all of the population VI thymocytes in HY male mice were phenotypically DN (**new Figure 4G**), and they expressed high levels of CD103 and PD-1 (**new Figure 4I**). These results align with DN TCR β^{hi} HY male thymocytes being derived from agonistic signaling-mediated negative selection.

To affirm this notion in another experimental model, we obtained AND TCR transgenic mice that were backcrossed from the H-2^b background to the selecting I-E^k background and that express the agonistic PCC peptide 88-104 [Oehen S. *et al.*, J Exp Med, PMID: 8676082] (kindly gifted by Dr. Xuguang Tai, CCR, NCI). In this thymic environment, the AND TCR becomes autoreactive. As shown below, the CD69 versus CCR7 plots of AND and AND I-E^k PCC⁺ mice were quite distinct, with a significant increase in negatively selected population VI cells of AND I-E^k PCC⁺ mice, concomitant with a decrease in population IV cells that are undergoing positive selection with increased TCR signaling intensity (**Figure 2 for Reviewer 1**). Collectively, these results from other autoreactive TCR transgenic mice further support our current findings.

Figure 2 for Reviewer 1.

A. CD69 versus CCR7 profiles of total thymocytes from AND and AND I-E^k PCC⁺ mice. Numbers indicate the frequency of cells in the corresponding gates whereby CD69⁺CCR7⁺ cells are population IV and CD69⁺CCR7^{int} cells are population VI.

B. Bar graph shows the frequency of population VI cells among total thymocytes of the indicated mice.

C. Histogram shows PD-1 expression on population IV thymocytes of AND and I-E^k PCC⁺ AND mice.

- The authors claim that “Thus, CD103 overexpression promotes negative selection in non-autoreactive TCR transgenes as well.” And “Collectively, these results showed that CD103 overexpression profoundly alters the development of thymocytes, and they further suggested that the clonal deletion of autoreactive thymocytes can be reinforced by CD103-mediated mechanisms.” How they reconcile the fact that CD103 overexpression promotes negative selection in mice that do not express the cognate antigen. These affirmations lack clear demonstrations of negative selection.

→ The reviewer raises a valid point that we wish to further elaborate on. Thymic positive selection is known to be mediated by weak TCR signaling, whereas negative selection is mediated by agonistic signaling. Also, positive selection is mediated by cTECs while negative

selection is thought to occur in the medulla. Our new results on the thymic architecture of 2D2 TCR mice demonstrate that the 2D2 thymus becomes highly disorganized by the overexpression of CD103, resulting in immature thymocytes gaining premature access to mTECs (**new Figures 6B, and 6D**), enhancing their binding to E-cadherin expressing mTECs, which would promote TCR signaling to reinforce negative selection. In this regard, we have assessed the frequency and phenotype of OT-II and OT-II.CD103^{Tg}β7^{Tg} population VI cells and found them to be substantially accumulated and contain a large fraction of PD-1 and Helios positive cells (**new Figures 4E and 4F**). Altogether, these results demonstrate that the forced expression of CD103 can mediate increased negative selection in support of our model.

-How the authors reconcile the interpretation of their results with an absence of CD103 expression in DN, DP, CD4 SP thymocytes and a strong expression in only CD8 SP thymocytes in WT mice (see Fig. 2a) ?

→ The selective expression of CD103 in post-selection CD8SP cells thymocytes is an intriguing observation that is controlled by the CD8-lineage specifying transcription factor, Runx3 [Grueter B. *et al.*, *J. Immunol.*, PMID: 16034110]. Our group has further expanded on this observation and recently reported a post-transcriptional mechanism of CD103 expression that requires the co-expression of and the heterodimerization with integrin β7 for CD103 cell surface expression [Keller HR., *et al.*, *Cell Mol Life Sci*, PMID: 34129058]. Moreover, we found that the limited availability of integrin β7 in pre-selection immature thymocytes constrains CD103 expression during thymic development, such that CD103 expression is mostly limited to post-selection thymocytes. To clarify these points, we have expanded our manuscript on these issues (lines 170-172).

- Mechanistically, the authors do not show how CD103 and CXCR4 are regulated.

→ The regulatory pathways of CD103 and CXCR4 expression have been previously mapped and were found to involve the transcription factors, Runx3 and E-proteins, respectively [Grueter B. *et al.*, *J. Immunol.*, PMID: 16034110; Kadakia T. *et al.*, *J Exp Med*, PMID: 31201207]. Whether 2D2 and other TCR transgenes would employ the same mechanisms to induce CD103 upon agonistic selection and to downregulate CXCR4 on DP thymocytes, however, remains currently unclear to us. In this regard, the reviewer is correct that we did not further explore the molecular mechanisms of CD103 and CXCR4 expression in our current study.

Nonetheless, we are happy to share that we made some promising observations that are now added to the revision. Prompted by our previous observations that the transcription factor RORγt is highly and specifically expressed in immature DP thymocytes and that RORγt controls cytokine receptor expression in these cells [Ligons DL. *et al.*, *Science Signaling*, PMID: 30154103], we assessed RORγt levels in DP thymocytes of different TCR transgenic mice to test a causal relationship with CXCR4 expression (**Figure 5F**). Unfortunately, we failed to observe any significant correlations between surface CXCR4 levels and RORγt expression, although we noticed a slight drop in RORγt levels upon the forced expression of CD103 (**revised Figure 5G**). Altogether, these results suggested that RORγt is unlikely to control CXCR4 expression downstream of transgenic TCRs. However, we found that the expression of Ikaros, which is another transcription factor that is highly upregulated in DP thymocytes [Mitchell JL. *et al.*, *Mol Immunol*, PMID: 28376432], was significantly induced in 2D2 TCR transgenic DP cells (**new Figure 5I**). More importantly, the genetic deletion of Ikaros in T cells resulted in increased CXCR4 expression in DP thymocytes (**new Figure 5J**), suggesting that Ikaros could be acting as

a transcriptional suppressor of CXCR4 expression. Thus, we consider it likely that 2D2 and other TCR transgenes could induce Ikaros expression on pre-selection DP thymocytes to downregulate CXCR4 expression. Whether this is indeed the regulatory mechanism of CXCR4 downregulation in 2D2 thymocytes requires further analyses, which we aim to address in our follow-up studies.

- They speculate that the “importance of thymocyte interaction with E-cadherin-expressing thymic epithelial cells (TECs)” without showing the expression of E-cadherin in mTEC. Does a specific subset of mTEC express E-cadherin?

→ These are two important and interesting issues that we now have addressed in the revision. First, we identified thymic epithelial cells in enzymatically digested thymic tissues using anti-EpCAM and anti-CD45 antibodies and assessed their E-cadherin expression. As shown in the **new Figure 5A**, E-cadherin was highly but also broadly expressed among TECs, indicating heterogeneity in E-cadherin expression among various TEC subsets.

Thus, we next aimed to map E-cadherin expression among different subsets of mTECs as requested by the reviewer. To this end, we reanalyzed public scRNA-Seq data of thymic epithelial cells for the expression of *Cdh1*, the gene encoding E-cadherin, in association with distinct markers of mTEC subsets, such as *Aire*, *Ccl21a*, but also *Trpm5*, which is associated with thymic Tuft cells (**new Figure 5B**). UMAP analyses identified 8 distinct populations, among which *Cdh1*-expressing cells were present in most clusters but specifically enriched in Tuft cells (cluster 2) and *Ccl21a*-expressing cells. Therefore, in agreement with the flow cytometry results, E-cadherin is evidently expressed in mTECs (**new Figure 5A**), but its abundance differs depending on the subset. We plan to follow up on the biological significance of these findings in our future studies.

-They authors conclude that 2D2 TCR promotes the generation of mature TCR^bhi CCR7⁺ MHC^I+ DN T cells, corresponding to negatively-selected cells without using reliable markers of negative selection such as cleaved caspase 3 (PMID: 31010850) or helios/PD1 markers (PMID: 23337809).

→ We agree that established markers of negative selection would be necessary to formally acknowledge TCR^β^{hi}CCR7⁺ DN thymocytes as negatively selected cells in 2D2 mice. To this end, we assessed surface PD-1, intracellular active caspase-3, and nuclear Helios expression as requested by the reviewer. Of note, in the revision, we were able to further refine the gating of negatively selected thymocytes by adding the surface marker CD69 for their identification, such that TCR^β^{hi} CCR7⁺ DN cells (in the original manuscript) are now TCR^β^{hi}CCR7^{int} CD69⁻ DN cells (also called population VI, henceforth). Positively selected CD4SP thymocytes, on the other hand, are identified as TCR^β^{hi}CCR7⁺CD69⁻ CD4SP cells (also called population V, henceforth). The modified gating strategies are shown in the **new Figures 3B, 3C, and 3D**, and the results for PD-1, active caspase-3, and Helios expression are shown in the **new Figures 3F, 3G, and 3E**, respectively. As expected, all markers associated with negative selection were highly induced in the negatively selected population VI but absent from positively selected population V cells, affirming that TCR^β^{hi}CCR7^{int}CD69⁻ DN thymocytes are the products of negative selection.

-Several conclusions such as “Examination of thymocyte maturation markers, i.e., CD5, CCR7, CD24, MHC-I, and CD69, indicated that 2D2 CD4SP cells were fully differentiated and comparable to WT CD4 T cells (Suppl. Fig. 1A)” and “Thus, the generation of CD4 T cells in

2D2 mice appeared to be intact despite their autoreactivity, and comparable to that of non-self-specific CD4 T cells in AND mice” are over-interpreted.

→ We thank the reviewer for pointing out these issues. We have carefully re-evaluated our manuscript and rephrased these conclusions to avoid overinterpretations of the results. The changes are made in lines 109-112 where we changed the original sentence to “....*surface markers that are associated with thymocyte selection and maturation, i.e., CD5, CCR7, CD24, MHC-I, and CD69, indicated that 2D2 CD4SP cells were comparable to WT CD4 T cells in their differentiation status.*”. We also changed the conclusion in lines 120-122 to “*Thus, the generation of CD4 T cells in 2D2 TCR transgenic mice appeared to be intact despite the autoreactivity of their TCR.*”

In many histograms, the MFI is not indicated and the expression is not quantified using several mice in several independent experiments.

→ In response to the reviewer’s request, we added the MFI to the histograms and quantified the expression. Specifically, we added the MFIs to the following figures; **revised Figures 1J and 1I, new Figures 5H, 5I, and 5J, and revised Figure 5K**, and the results were quantified in the associated bar graphs.

In Suppl. Fig. 1A, the authors should analyze the maturation status of 2D2 CD4⁺ SP thymocytes based on CD69/MHCI expression (PMID: 27043411).

→ This is an excellent point. As suggested, we employed the distinct expression of CD69 versus MHC-I among CCR7⁺TCRβ⁺ post-selection thymocytes to analyze the maturation status of 2D2 CD4SP thymocytes. The results are added as **new Suppl. Figure 3A**, and they demonstrate that 2D2 CD4SP cells are mostly comprised of fully mature T cells.

- I strongly encourage the authors to show all the data indicated as “not shown” and in particular those related to the following sentence “we next assessed the expression of a series of developmental and differentiation markers by reanalyzing public RNAseq data sets of agonistic signaled thymocytes and screening custom-made panels of antibody arrays for surface molecules (data not shown)” justifying the analysis of CD103 in negative selection.

→ As suggested, we removed all “data not shown” from the revised manuscript, such that now we only explain and discuss results displayed in the manuscript. In this regard, we added the results from screening surface molecules that are induced on negatively selected TCRβ^{hi} DN but not on positively selected TCRβ^{hi} CD4SP cells in 2D2 mice to the revision (**new Suppl Figures 1D and 1E**). These molecules include a series of cytokine receptors, activation markers, and integrins, and it is through this screening process that we identified the marked upregulation of integrin CD103 in TCRβ^{hi} DN thymocytes (**revised Figure 1I**).

-The interpretation of results shown in Fig 4C should be reinforced by analyzing the localization of mTEC in these transgenic mice. It seems that the cortico-medullary demarcation is totally blurry in 2D2.CD103Tgβ7Tg. The authors should quantify medullary areas in 2D2 and 2D2.CD103Tgβ7Tg mice.

→ The reviewer is correct to point out the blurred cortico-medullary junction (CMJ) in 2D2 mice, which we found to be further exacerbated in 2D2.CD103^{Tg}β7^{Tg} mice. By retaking the H&E histochemistry images (**revised Figure 6A**), we affirm that the blurriness is not due to low resolution of the images but due to fragmentation of the medulla and dispersion of the CMJ. For

an independent evaluation of our findings, we initiated a new collaboration with Dr. Yousuke Takahama (National Cancer Institute, NIH), who helped us to map the mTEC and cTEC locations by immunohistochemistry, affirming the altered thymic architecture in 2D2 TCR transgenic mice (**new Figure 6C**). Also, as requested, we quantified the medullary areas of the 2D2 and 2D2.CD103^{Tg}β7^{Tg} mice, for which we employed the help of Dr. Michael Kruhlak (CCR Confocal Microscopy facility, NCI, NIH). We quantified the medullary and cortical areas using Arivis Pro software (Zeiss) and then applied machine learning pixel classifiers (random forest algorithm) to determine the ratio of medulla to cortex. The results revealed a dramatically reduced medulla to cortex ratio in 2D2.CD103^{Tg}β7^{Tg} mice, demonstrating the detrimental effect of CD103 overexpression in thymic development (**new Figure 6D and Materials and Methods**).

-The CD4/CD8 profile of 2D2 mice is not consistent in panels Fig. 1a and Fig.2d.

→ We acknowledge that the CD4 versus CD8 profiles of 2D2 mice were inconsistent, and we have replaced them with correctly representative data, as shown in **revised Figure 1A** and **Figure 2D**.

-Consider using an alternative approach to the t-tests that are used for some comparisons of more than two groups

→ We appreciate the suggestion. Following the reviewer's suggestion, we tested the statistical power using one-way ANOVA for the **Figures 1B, 1C, 1G, Figures 2C, 2D, 2E, Figures 3A, 3C, and Figures 5C, 5E, 5G, 5H, 5I, 5K** and **Figures 7A, 7B, 7C, and 7D**, in the revision

- statistics between WT and 2D2 are missing (Fig. 1B, C).

→ As requested, we added the statistics of cell numbers and frequencies between WT and 2D2 mice to the bar graphs in **the revised Figures 1B** and **1C**.

Reviewer #2 (Remarks to the Author):

Liman et. al. propose that CD103 expression is part of a developmental pathway for autoreactive thymocytes. Specifically, the authors propose that negative selection of MHCII-restricted thymocytes is associated with the induction of CD103 integrin. Further, overexpression of CD103 suppressed CXCR4 expression, which promoted clonal deletion. The authors studied 2D2 TCR transgenic mice, whereby T cells are specific to a MOG peptide, and are used as a model of neuritis/EAE caused by self-reactive T cell clones. In these mice, the currently accepted view is that self-reactive thymocytes escape negative selection because there is a low level of expression of the negative-selecting antigen in the thymus. In other words, developing thymocytes fail to encounter their negative-selecting antigen, are therefore not deleted, and leave the thymus as self-reactive T cells. The authors propose that instead, self-reactive 2D2 thymocytes undergoing agonist selection divert negative selection in a process that involves expression of CD103. This integrin promotes thymocyte interaction with E-cadherin expressed in medullary thymic epithelial cells. The authors show that the overall cortical/medullary areas are altered in 2D2 mice, and even more so in 2D2 overexpressing integrin CD103. The data is overall interesting, the models selected are informative and analyses are appropriate.

→ We are grateful for the expert review of our study and for teasing out the core issues of our manuscript. Prompted by the reviewer's comments, we discussed our current findings in the context of the prevailing view in further detail, which we believe helped significantly in improving our revision. We appreciate the reviewer's constructive suggestions.

There are, however, aspects in the manuscript that require additional attention. TCR transgenic mice can be useful to decipher specific aspects of T cell differentiation. Nevertheless, they often differ from wild type mice in the time at which they express their TCR, and that can affect the normal differentiation of those cells. It is important to clarify whether CD103 is relevant for negative selection in wild type mice, or to which extent it is a specificity of this transgenic TCR mouse line.

→ We concur that data from TCR transgenic mouse models can sometimes conflict with the results of non-TCR transgenic WT mice. This is also why we employed two additional TCR transgenic mouse strains (e.g., AND and OT-II) in our original submission to replicate our observations in other TCR transgene models (**Figures 4A-4D, and new Figures 4E and 4F**). Nonetheless, these results cannot exclude TCR transgene effects as the reviewer correctly points out.

Thus, in response to the reviewer's request to assess the relevance of CD103 in WT mice, we obtained CD103-deficient mice (*Itgae^{-/-}*) and examined whether the lack of CD103 is associated with a decrease in negatively selected thymocytes. Based on our refined thymocyte analysis using the markers CD69 and CCR7 (**new Figures 3B and 3C**), we matched negatively selected DN TCRβ^{hi} cells to a newly defined population of CD69⁻CCR7^{int} cells that we refer to as "population VI (pop. VI)" (**new Figure 3C**). Total thymocytes can be assigned into five distinct developmental stages based on CD69 and CCR7 expression that visualize positive selection and maturation of T cells in the thymus [Kadakia T *et al.*, J Exp Med, PMID: 31201207]. Using the same markers, we were able to identify thymocytes that underwent negative selection to reside among a new subset of CD69-negative (CD69⁻) CCR7^{int} cells (*i.e.*, pop. VI), as shown in **Figure 1 for the Reviewer 2** (see below) and in **new Figure 3C** of the revised manuscript. Notably, the fraction of population VI is easily discernable among WT

thymocytes and, as expected, dramatically increased in 2D2 thymocytes (**new Figure 3C**). Moreover, population VI cells express high levels of PD-1 (**new Figure 3F**) and exert substantially increased caspase-3 activity (**new Figure 3G**), affirming them to result from negative selection.

Figure 1 for Reviewer 2. Total thymocytes of WT and 2D2 mice were plotted for CD69 versus CCR7 expression, revealing 6 distinct populations of developmental stages. Stages 1-5 had been previously described [PMID: 31201207]. In the current study, we identified a new population (pop. VI; marked in red) that mostly corresponds to agonistically signaled, negatively selected cells that are dramatically enriched in 2D2 TCR thymocytes.

In response to the reviewer's request, we then analyzed thymocytes of CD103-deficient mice to assess the generation of negatively selected cells. Here, we found that *Itage*^{-/-} thymocytes were significantly decreased in their frequency of population VI cells (**new Figure 3H**), agreeing with the role of CD103 in promoting negative selection. Since these new data are generated with TCR non-transgenic mice, we hope that the reviewer will agree that the CD103 effect is independent of the timing or specificity of the transgenic TCR expression.

Specific comments:

Lines 119 to 122 and 137 to 139: Description of DN cells should include the normal DN stages using at least CD44 and CD25 to show when the Tg TCR starts to be expressed. That could help clarifying the stage from which differentiation might be altered. This can be important for data interpretation at the end of the same paragraph, where the authors state that mature DN bTCRhi result from negative selection. While the authors cite 2 papers to state that DN bTCRhi result from DP undergoing negative selection, the papers cited refer to CD8αα TCRαβ precursors in the thymus, and the authors do not formally show that the same happens in the 2D2 thymocytes. I consider important to show experimentally that: 1) DN bTCRhi indeed result from DP and 2) bTCR expression in the normal DN stages of differentiation (DN1 to DN4).

→ These are both excellent points.

Regarding the request to show TCRβ expression in distinct stages of DN thymocyte differentiation, we used the classical markers CD25 and CD44 to identify DN1–DN4 subsets as suggested by the reviewer. In contrast to conventional DN1–DN4 analyses, however, we did not add anti-TCRβ antibodies into our lineage-negative (Lin⁻) dump gate when excluding lineage-committed cells from our DN thymocyte analysis. We only used antibodies against CD8α, B220, NK1.1, CD11b, Ly6G/Ly6C, and γδ TCR to exclude lineage marker-expressing cells such that the timing of the transgenic TCR expression can be assessed (**new Suppl. Figure 3C**). Due to this limitation, we would like to advise the reviewer that the omission of anti-TCRβ antibodies makes it challenging to directly compare our CD25 versus CD44 subset analysis to the conventional profile. Nonetheless, we found that transgenic 2D2 TCR expression already started in the earliest thymic progenitors (DN1 cells), as we detected both TCRβ and clonotypic TCR Vα3.2 expression in DN1 thymocytes of 2D2 mice (**new Suppl. Figure 3D**).

Regarding the question of whether DN TCRβ^{hi} cells result from DP thymocytes, unfortunately, we are unable to provide a straightforward answer at this point. Nonetheless, we tried to address this issue with the reagents at hand, and we tested this question using transgenic

mice expressing the Cre recombinase under the control of CD8 E8iii promoter/enhancer that is specifically activated in immature DP thymocytes [Park JH *et al.* Nat Immunol, 2010, PMID: 20118929]. By breeding E8iii Cre transgenic mice with floxed gene reporter mice, it is possible to monitor the developmental pathway of immature thymocytes. The deletion of the reporter protein will indicate that the DP stage-specific Cre had been activated, and that the cells have passed through the DP stage. The IL-7R α is a cytokine receptor that is highly expressed on mature thymocytes, including CD4SP and negatively selected PD-1⁺ DN thymocytes that are both TCR β^{hi} (**Figure 2 for Reviewer 2**).

Figure 2 for Reviewer 2. CD4SP thymocytes and PD-1⁺ DN thymocytes of *Il7r^{fl/fl}.WT* (blue) and *Il7r^{fl/fl}.E8iii.Cre⁺* (orange) mice. The expression of IL-7R α is reduced on CD4SP thymocytes and PD-1⁺ DN thymocytes upon activation of E8iii.Cre⁺ at DP thymocyte stage.

When assessing IL-7R α expression on PD-1⁺ DN cells of IL-7 receptor floxed E8iii.Cre mice (*Il7r^{fl/fl}.E8iii.Cre⁺*), we found them to have completely lost IL-7R α expression, strongly suggesting that these post-selection thymocytes are derived from DP cells. While not formally establishing a precursor-progeny relationship

between DP and post-selection DN TCR β^{hi} cells, these results are clearly in support of this model, and we hope that the reviewer will agree.

Lines 192 to 195: interpretation lack formal demonstration that those thymocytes result from negative selection.

→ Here, the reviewer is referring to our statement that we consider “CXCR4⁻ CCR7^{intb}” cells as negatively selected thymocytes (lines 192 – 195 of the original manuscript). In hindsight, we realize that this statement appears as overinterpretation of our data, and we agree with the reviewer that we should establish formal demonstration of our proposition.

Negative selection is necessarily associated with strong agonistic TCR signaling, which can be visualized by the induction of PD-1 and expression of the transcription factor Helios, among others [Kreslavsky T *et al.*, J Exp Med, PMID: 23980099; Ross EM *et al.*, Eur J Immunol, PMID: 24740292; Daley R *et al.*, J Exp Med, PMID: 23337809]. Thus, we employed these established markers of thymic negative selection to reassess the phenotypes of CXCR4⁻ CCR7^{int} thymocytes. Of note, we found that CD69 versus CCR7 staining can identify negatively selected cells with higher accuracy than CXCR4 versus CCR7, which we now refer to as “population VI” within the CD69 versus CCR7 plot (**new Figure 3C**). Accordingly, we applied these markers to reanalyze negative selection (**new Figure 3C**), where we found that “CXCR4⁻ CCR7^{intb}” cells, and more specifically “CD69⁻ CCR7^{intb}” (population VI) thymocytes express large amounts of PD-1 and high levels of Helios (**new Figures 3E and 3F**). These results strongly support our thesis that “CD69⁻ CCR7^{intb}” cells are the products of negative selection.

Moreover, it is well established that negatively selected thymocytes become prone to apoptosis, and that apoptotic cells can be quantitated by Caspase-3 activation [Kreslavsky T *et al.*, J Exp Med, PMID: 23980099]. Notably, we found that 2D2 transgenic DN thymocytes in population VI contained dramatically increased amounts of active Caspase-3 compared to positively selected CD4SP 2D2 transgenic cells (**new Figure 3G**). Altogether, these results are in support of attributing “CD69⁻ CCR7^{intb}” (population VI) cells as the products of negative selection.

Lines 228 to 230: the statement “Thus, immature 2D2 thymocytes are equipped to engage mTECs because they reside in the correct anatomical location.” Requires formal demonstration. Immunohistological analyses with the appropriate markers are required to show what the authors state.

→ Upon re-reading our statement in lines 228-230, we realize that this could be easily considered an overinterpretation of the results. Hence, we removed this sentence and dampened down our assessment such that the revised statement reads as follows: “*Thus, immature 2D2 thymocytes could gain premature access to mTECs, and the overexpression of CD103 would promote their binding to and negative selection by mTECs.*”. Here, we wish to emphasize that alteration in intrathymic trafficking can disturb the proper thymic selection and anatomical distribution of immature thymocytes.

To visualize this effect and in response to the reviewer’s criticism, we initiated a new collaboration with Dr. Yousuke Takahama’s lab (National Cancer Institute, NIH) to obtain expert opinion and data on the thymic architecture of 2D2 and 2D2.CD103^{Tg}β7^{Tg} mice. We prepared new high-resolution images of our H&E histochemistry analysis, demonstrating a diffused corticomedullary junction in 2D2 mice (**revised Figure 6A**) that was associated with fragmented medullary regions (**new Suppl. Figure 4C**). We further performed immunohistochemistry of thymic sections with antibodies specific to cTECs, *i.e.*, CD205 (DEC-205) or Ly51, and specific to mTECs, *i.e.*, Keratin 14 (K14) or UEA-1, to visualize the marked differences in thymic architecture of 2D2 mice compared to WT controls (**new Figure 6C**). Because of the blurred distinction between the cortex and the medulla as well as the highly fractionated medullary structure in 2D2 thymocytes, these results support our notion that 2D2 thymocytes are exposed to a drastically altered thymic microenvironment, which is further exacerbated in 2D2.CD103^{Tg}β7^{Tg} mice (**new Suppl. Figure 4D**).

Lines 235 to 237: Sections depicted in Fig. 4C, Suppl. Fig. 4 are not clear and, by far, insufficient to support the author’s claims. Medullary areas should be shown and DN thymocytes that might be present in the medullas should be quantified. This is essential, especially if the authors want to keep their conclusion “that alters the intrathymic trafficking and distribution of preselection thymocytes which in turn can make them susceptible to increased negative selection upon CD103 overexpression” (in lines 262 to 264).

→ In line 237 of the original manuscript, we noted “a broad dispersion of thymocytes throughout the thymus” that we proposed to be indicative of immature thymocytes gaining premature access to mTECs in 2D2 mice. In support of this notion, we previously generated H&E-stained thymic sections of 2D2 TCR transgenic mice (**Figure 4C** and **Suppl. Figure 4** of the original submission), showing that the corticomedullary junction has become blurry with a concomitant decrease in the medullary area and fragmentation of the medulla. This effect was exacerbated by the CD103 transgene, which prompted us to suggest that it would lead to “increased negative selection upon CD103 overexpression”.

Upon re-examining these figures, however, we admit that the image quality of the histology is sub-par and that the lack of quantification makes it difficult to interpret the results convincingly. Thus, following the reviewer’s suggestion, we recaptured the images, quantified the results, and performed immunohistochemistry for better visualization of the mTEC and cTEC areas (**new Figures 6B, 6C, and 6D**). Specifically, we collaborated with Dr. Jan Wisniewski (NCI) to generate high-quality histochemistry images to discern the thymic cortex and medulla

of the experimental mice (**revised Figure 6A**) and quantified the relative areas using in-house image mapping programs (**new Figure 6B and Materials and Methods**). Because the medullary area in 2D2.CD103^{Tg}β7^{Tg} mice were extremely fragmented and difficult to discern with this method, we then collaborated with Dr. Michael Kruhlak's CCR Confocal Microscopy facility (NCI, NIH) and used machine learning software to determine the ratio of medulla to cortex, which revealed a dramatically reduced ratio in 2D2.CD103^{Tg}β7^{Tg} mice (**new Figure 6D and Materials and Methods**).

Minor comments:

Lines 102, 103: where the authors describe the similarities between the wt and the 2D2 thymocytes (Fig S1), also the differences should be stated.

→ In showing **Suppl. Figure S1**, we originally aimed to demonstrate that CD4SP thymocytes of 2D2 mice are developmentally mature. To this end, we assessed multiple surface markers that are highly expressed on post-selection mature thymocytes, such as CD5, CCR7, MHC-I, and CD69, and showed that 2D2 CD4SP cells express large amounts of these markers, affirming them as post-selection thymocytes. On the other hand, the reviewer is correct that careful examination reveals that 2D2 CD4SP cells express lower levels of CD5 and CD69 compared to their WT counterparts. We hypothesize that the 2D2 TCR is a low-affinity receptor, which may explain why monoclonal 2D2 CD4SP cells express lower levels of both CD5 and CD69 compared to the polyclonal WT CD4SP population. This suggests that the 2D2 TCR is undergoing low-affinity tonic signaling. We are now elaborating on this point in the revised manuscript, lines 112-115.

Fig. 1H is not described in the text.

→ We wish to draw the reviewer's attention to lines 134-138 (lines 123-126 in the original manuscript), where Figure 1H has been described. Thus, we will consider this issue to be resolved.

Lines 156 to 158: for clarity, please correct sentence to avoid "introduced" the transgenes. I would favor "crossed" the lines to generate...

→ As requested, we removed "introduced" from the manuscript and replaced it with "crossbred" throughout the manuscript. These changes were made in lines 168 and 240.

Fig. 4A: specify "DP thymocytes" in the figure

→ Figure 4A is now **the revised Figure 5C** in the revision. We had previously labeled the graph as "immDP" cells, which is indeed confusing. We have now spelled out this abbreviation, and it reads "immature DP thymocytes".

Line 266: If cells do not exit the thymus, they don't get "depleted" in the periphery. This subtitle should be corrected for clarity.

→ We concur, and to avoid overinterpretation of our results, we have changed the subtitle to "CD103 overexpression is associated with a decrease in peripheral 2D2 DN T cells" (line 351).

REVIEWERS' COMMENTS

Reviewer #1 (Remarks to the Author):

I appreciate the comprehensive revisions made by the authors in response to my initial concerns about the descriptive nature of the study and the lack of mechanistic insights. The new data presented in this revised version substantially reinforce the conclusions of the manuscript. In particular, the use of other autoreactive TCR transgenic mice (HY and AND TCR transgenic mice) provides stronger support for the authors' conclusions. More importantly, the use of TCR non-transgenic CD103-deficient mice to examine negative selection provides more physiologically relevant evidence to support a role for CD103 in thymic selection. In this regard, the authors should highlight this point more clearly in the abstract. The new gating strategy based on CD69 and CCR7 is also more convincing in identifying negatively selected thymocytes.

Overall, these additions substantially improve the physiological relevance and novelty of the findings.

→ We thank the reviewer for the succinct summary of our revised study, and for the positive comments on our revision.

Minor points:

-I encourage the authors to show an H&E counterstaining in Figure 6A for the thymus of 2D2.CD103Tg.b7Tg mice at the same magnification as for the WT and 2D2 mice, given that the thymus of these mice is quantified in Figure 6B.

→ Of note, the quantitative analysis in Figure 6B is based on the histological images provided in Suppl. Fig. 4C, and not Figure 6A. Because Suppl. Fig. 4C already contains the thymus images of all three mice, *i.e.*, WT, 2D2, and 2D2.CD103^{Tg}β7^{Tg}, we consider the reviewer's comment on Figure 6B to be fully addressed in the revision.

-Statistical analysis is missing in Figs 7H and 7I.

→ The statistical significance in Figure 7H was originally placed on the upper left-hand side of the graph. We realized that it can be easily missed, such that we have revised Figure 7H, now clearly indicating the statistical analysis on right side of the graph.

Regarding Figure 7I, it is the journal policy not to derive statistics from experiments with $n < 3$ replicates. Since Figure 7I represents the cumulative disease incidence of two independent EAE experiments, we did not draw a statistical conclusion in accordance with the journal's instructions.

-Mean Fluorescence Intensity (MFI) values are not indicated in the histograms of several figures (Figs. 2A, 2B, 3E, 3F, 5D, and 5F).

→ In response to the reviewer's request, we added the MFI values to the histograms as shown in modified Figures 2A, 2B, 3E, 3F, 5D, and 5F.

Altogether, we hope that these changes have resolved any outstanding issues, and we thank the reviewer again for all the constructive feedback and suggestions.

Reviewer #2 (Remarks to the Author):

The authors considered and presented additional experiments that replied to all my comments and concerns. They indeed went well beyond the originally submitted data. I consider that the paper is very nice and recommend it to be accepted for publication.

→ We greatly appreciate the reviewer's comments on our revision, and for mediating comments for reviewer #1. The reviewer carefully checked each of our response and found them to have sufficiently addressed all concerns. We thank the reviewer again for endorsing this manuscript.

#####

(Reviewer #2 was asked by the editor to mediate for reviewer #1, but later on reviewer #1 submitted their own report. After concurred by reviewer #2, the editor is posting below the mediating comment as a reference:)

#####

Mediating comments to author responses to R#1:

I consider that the substantial amount of new data provided in this revised version plus the corrections in the text fully address all concerns raised by Reviewer #1. I went through every single point raised, and consider that the revisions address every point. Finally, while the reviewer was indeed very critical of the manuscript, his/her remarks contributed to a better manuscript, that goes beyond the originally submitted version. I congratulate the authors for their work.

Below, you may find my point-by-point assessment, where my comment for every point raised is the 3rd, following the authors' reply.

The study by Liman et al. attempts to investigate how autoreactive MHCII-restricted thymocytes evade negative selection. They propose that negative selection is associated with the integrin CD103 and show that the overexpression of CD103 suppresses CXCR4 expression and reinforces clonal deletion. In summary, although the authors suggest a new role of CD103 in negative selection, the relevance of their results, which are based on artificial systems, is questionable.

→ We thank the reviewer for the succinct review and constructive criticisms. We greatly appreciate the comments, but we also respectfully disagree that the results from TCR transgenic mouse models would be physiologically irrelevant. In fact, the 2D2 TCR transgenic mice have spawned hundreds of publications (based on a PubMed search for "2D2" on April 4, 2025), helping us to understand how autoimmunity is triggered and can be resolved. As such, we remain confident that employing genetically engineered mouse models to study fundamental mechanisms in immunology remains highly relevant.

This point is an opinion. The authors use a transgenic system to learn about the role of CD103 in negative selection. I consider that the point is well discussed in the paper and the limitations discussed.

In my opinion, this study is overall descriptive and suffers from a lack of clear demonstrations and mechanistic insights. At this stage the main results are not sufficiently convincing to represent a major breakthrough in the field and justify publication in Nature Communications. →We understand that the reviewer is concerned about the mechanistic insight and the broader impact of our findings. Prompted by the reviewer's comments, we now have performed further in-depth studies into the effect of CD103 in thymopoiesis using new mouse models and new experimental tools. Among others, we 1) examined the role of CD103 in negative selection using TCR non-transgenic CD103-deficient mice, 2) assessed E-cadherin expression on thymic epithelial cells (TECs) and mapped its distribution among medullary TECs (mTECs) by reanalyzing scRNA-Seq data, 3) initiated new collaborations to visualize and quantify changes in the thymic architecture by 2D2 and CD103 expression, and 4) performed adoptive transfer experiments and employed the Experimental Autoimmune Encephalomyelitis (EAE) model to demonstrate the physiological significance of CD103 expression or its absence in CD4 T cells. Most importantly, we further refined the phenotype of DN thymocytes that have resulted from agonistic negative selection (Figure 1 for Reviewer 1).

In our original submission, we had used the CXCR4 versus CCR7 plots to identify negatively selected DN thymocytes (Figure 1 for Reviewer 1, top). During the revision, we found that assessing CD69 versus CCR7 expression of total thymocytes permits a much more precise identification of these cells that we found being highly enriched in population VI (Figure 1 for Reviewer 1, bottom). Consequently, we employed this new gating strategy to characterize the role of CD103 in the negative selection of 2D2 TCR transgenic mice in the revision, helping us to gain further insights into the mechanistic aspects of our findings.

I consider that the revised version addresses the concerns of the reviewer.

Specific comments:

-As stated by the authors "How to evade negative selection, however, remains incompletely understood" is an important question. However, this study does not answer to this question, but rather shows that CD103 "reinforces clonal deletion".

→We agree that our question is inadequately formulated. We further appreciate the reviewer's comment that our study rather addresses a role for integrin CD103 in reinforcing thymic negative selection, such that this statement is misleading. We fixed this issue by rephrasing this statement and amended the revised manuscript accordingly (lines 39-41).

I consider that the revised version addresses the concerns of the reviewer.

-It is surprising that all the conclusions are based on CD103 overexpression and that the authors didn't use a conditional Ko of CD103 in thymocytes, to support their conclusions or at least compare their results of overexpression with those of CD103 ko mice.

→To address the issue of not having tested the effects of CD103 deficiency, we obtained CD103-deficient mice and analyzed their thymocyte development. Of note, CD103 conditional KO mice had not been reported and are not available. However, the reviewer's comment is very much appreciated, and specific deletion of CD103 in immature thymocytes would be the ideal model to test the role of CD103 in thymic selection. Nonetheless, we consider generating conditional CD103-KO mice to be outside the scope of our current study, and we hope that the new results from CD103 germline KO mice will suffice to answer the reviewer's question (new Figure 3H). Specifically, we assessed CD103-KO thymocytes for the presence of population VI cells in the CD69 versus CCR7 plot, where we found them to be markedly reduced compared to WT

thymocytes (new Figure 3H). These results are in full agreement with our model where CD103 reinforces negative selection in the thymus.

I consider that the revised version fully addresses the concerns of the reviewer.

-It would be important to define and show the impact of CD103 overexpression or deletion in a non-TCR transgenic system.

→We agree with the reviewer, and we have now assessed the impact of CD103-deficiency in a non-TCR transgenic model. To this end, we obtained mice that are deficient in *Itgae*, the gene encoding CD103, and assessed the appearance of negatively selected DN thymocytes in comparison to those of WT mice. As mentioned above, we found that the CD69 versus CCR7 plot provides a higher resolution and more accurate identification of negatively selected DN thymocytes than our previous CXCR4 versus CCR7 plot (Figure 3A, new Figures 3B and 3C). Thus, we used the gating strategy based on CD69 and CCR7 to identify thymocytes undergoing negative selection and found a significant reduction of Population VI cells in CD103-deficient mice (new Figure 3H). These results agree with our model that CD103 expression reinforces negative selection in the thymus.

I consider that the revised version fully addresses the concerns of the reviewer.

- It would also be important to use other autoreactive TCR transgenic mice to support their findings.

→This is an excellent point, as we would indeed expect to observe the same results in other autoreactive TCR models. To this end, we acquired HY TCR transgenic mice whose TCR is specific to a male antigen [Kisielow P. et al., *Nature*, PMID: 3260350]. Consequently, HY TCR thymocytes are autoreactive in male mice. Strikingly, we found that the overwhelming majority of thymocytes in HY male mice resided within the population VI gate, which corresponds exactly to the region where we would expect to find negatively selected cells (new Figures 4G and 4H). Moreover, all of the population VI thymocytes in HY male mice were phenotypically DN (new Figure 4G), and they expressed high levels of CD103 and PD-1 (new Figure 4I). These results align with DN TCR β hi HY male thymocytes being derived from agonistic signaling-mediated negative selection.

To affirm this notion in another experimental model, we obtained AND TCR transgenic mice that were backcrossed from the H-2b background to the selecting I-Ek background and that express the agonistic PCC peptide 88-104 [Oehen S. et al., *J Exp Med*, PMID: 8676082] (kindly gifted by Dr. Xuguang Tai, CCR, NCI). In this thymic environment, the AND TCR becomes autoreactive. As shown below, the CD69 versus CCR7 plots of AND and AND I-Ek PCC+ mice were quite distinct, with a significant increase in negatively selected population VI cells of AND I-Ek PCC+ mice, concomitant with a decrease in population IV cells that are undergoing positive selection with increased TCR signaling intensity (Figure 2 for Reviewer 1). Collectively, these results from other autoreactive TCR transgenic mice further support our current findings.

I consider that the revised version fully addresses the concerns of the reviewer.

- The authors claim that “Thus, CD103 overexpression promotes negative selection in non-autoreactive TCR transgenes as well.” And “Collectively, these results showed that CD103 overexpression profoundly alters the development of thymocytes, and they further suggested that the clonal deletion of autoreactive thymocytes can be reinforced by CD103-mediated mechanisms.” How they reconcile the fact that CD103 overexpression promotes negative

selection in mice that do not express the cognate antigen. These affirmations lack clear demonstrations of negative selection.

→The reviewer raises a valid point that we wish to further elaborate on. Thymic positive selection is known to be mediated by weak TCR signaling, whereas negative selection is mediated by agonistic signaling. Also, positive selection is mediated by cTECs while negative selection is thought to occur in the medulla. Our new results on the thymic architecture of 2D2 TCR mice demonstrate that the 2D2 thymus becomes highly disorganized by the overexpression of CD103, resulting in immature thymocytes gaining premature access to mTECs (new Figures 6B, and 6D), enhancing their binding to E-cadherin expressing mTECs, which would promote TCR signaling to reinforce negative selection. In this regard, we have assessed the frequency and phenotype of OT-II and OT-II.CD103Tg β 7Tg population VI cells and found them to be substantially accumulated and contain a large fraction of PD-1 and Helios positive cells (new Figures 4E and 4F). Altogether, these results demonstrate that the forced expression of CD103 can mediate increased negative selection in support of our model.

Demonstration of the effect on negative selection is difficult but I consider that the authors addressed the concerns of the reviewer.

-How the authors reconcile the interpretation of their results with an absence of CD103 expression in DN, DP, CD4 SP thymocytes and a strong expression in only CD8 SP thymocytes in WT mice (see Fig. 2a) ?

→The selective expression of CD103 in post-selection CD8SP cells thymocytes is an intriguing observation that is controlled by the CD8-lineage specifying transcription factor, Runx3 [Grueter B. et al., J. Immunol., PMID: 16034110]. Our group has further expanded on this observation and recently reported a post-transcriptional mechanism of CD103 expression that requires the co-expression of and the heterodimerization with integrin β 7 for CD103 cell surface expression [Keller HR., et al., Cell Mol Life Sci, PMID: 34129058]. Moreover, we found that the limited availability of integrin β 7 in pre-selection immature thymocytes constrains CD103 expression during thymic development, such that CD103 expression is mostly limited to post-selection thymocytes. To clarify these points, we have expanded our manuscript on these issues (lines 170-172).

I consider the answer of the authors to be sufficient to address the comment.

- Mechanistically, the authors do not show how CD103 and CXCR4 are regulated.

→The regulatory pathways of CD103 and CXCR4 expression have been previously mapped and were found to involve the transcription factors, Runx3 and E-proteins, respectively [Grueter B. et al., J. Immunol., PMID: 16034110; Kadakia T. et al., J Exp Med, PMID: 31201207]. Whether 2D2 and other TCR transgenes would employ the same mechanisms to induce CD103 upon agonistic selection and to downregulate CXCR4 on DP thymocytes, however, remains currently unclear to us. In this regard, the reviewer is correct that we did not further explore the molecular mechanisms of CD103 and CXCR4 expression in our current study.

Nonetheless, we are happy to share that we made some promising observations that are now added to the revision. Prompted by our previous observations that the transcription factor ROR γ t is highly and specifically expressed in immature DP thymocytes and that ROR γ t controls cytokine receptor expression in these cells [Ligons DL. et al., Science Signaling, PMID: 30154103], we assessed ROR γ t levels in DP thymocytes of different TCR transgenic mice to test a causal relationship with CXCR4 expression (Figure 5F). Unfortunately, we failed to observe

any significant correlations between surface CXCR4 levels and ROR γ t expression, although we noticed a slight drop in ROR γ t levels upon the forced expression of CD103 (revised Figure 5G). Altogether, these results suggested that ROR γ t is unlikely to control CXCR4 expression downstream of transgenic TCRs. However, we found that the expression of Ikaros, which is another transcription factor that is highly upregulated in DP thymocytes [Mitchell JL. et al., Mol Immunol, PMID: 28376432], was significantly induced in 2D2 TCR transgenic DP cells (new Figure 5I). More importantly, the genetic deletion of Ikaros in T cells resulted in increased CXCR4 expression in DP thymocytes (new Figure 5J), suggesting that Ikaros could be acting as a transcriptional suppressor of CXCR4 expression. Thus, we consider it likely that 2D2 and other TCR transgenes could induce Ikaros expression on pre-selection DP thymocytes to downregulate CXCR4 expression. Whether this is indeed the regulatory mechanism of CXCR4 downregulation in 2D2 thymocytes requires further analyses, which we aim to address in our follow-up studies. The revised version fully addresses the concerns of the reviewer.

- They speculate that the “importance of thymocyte interaction with E-cadherin-expressing thymic epithelial cells (TECs)” without showing the expression of E-cadherin in mTEC. Does a specific subset of mTEC express E-cadherin?

→ These are two important and interesting issues that we now have addressed in the revision. First, we identified thymic epithelial cells in enzymatically digested thymic tissues using anti-EpCAM and anti-CD45 antibodies and assessed their E-cadherin expression. As shown in the new Figure 5A, E-cadherin was highly but also broadly expressed among TECs, indicating heterogeneity in E-cadherin expression among various TEC subsets. Thus, we next aimed to map E-cadherin expression among different subsets of mTECs as requested by the reviewer. To this end, we reanalyzed public scRNA-Seq data of thymic epithelial cells for the expression of *Cdh1*, the gene encoding E-cadherin, in association with distinct markers of mTEC subsets, such as *Aire*, *Ccl21a*, but also *Trpm5*, which is associated with thymic Tuft cells (new Figure 5B). UMAP analyses identified 8 distinct populations, among which *Cdh1*-expressing cells were present in most clusters but specifically enriched in Tufts cells (cluster 2) and *Ccl21a*-expressing cells. Therefore, in agreement with the flow cytometry results, E-cadherin is evidently expressed in mTECs (new Figure 5A), but its abundance differs depending on the subset. We plan to follow up on the biological significance of these findings in our future studies.

The revised version fully addresses the concerns of the reviewer.

-They authors conclude that 2D2 TCR promotes the generation of mature TCR β hi CCR7+ MHC1+ DN T cells, corresponding to negatively-selected cells without using reliable markers of negative selection such as cleaved caspase 3 (PMID: 31010850) or helios/PD1 markers (PMID: 23337809).

→ We agree that established markers of negative selection would be necessary to formally acknowledge TCR β hiCCR7+ DN thymocytes as negatively selected cells in 2D2 mice. To this end, we assessed surface PD-1, intracellular active caspase-3, and nuclear Helios expression as requested by the reviewer. Of note, in the revision, we were able to further refine the gating of negatively selected thymocytes by adding the surface marker CD69 for their identification, such that TCR β hi CCR7+ DN cells (in the original manuscript) are now TCR β hiCCR7int CD69– DN cells (also called population VI, henceforth). Positively selected CD4SP thymocytes, on the other hand, are identified as TCR β hiCCR7+CD69– CD4SP cells (also called population V,

henceforth). The modified gating strategies are shown in the new Figures 3B, 3C, and 3D, and the results for PD-1, active caspase-3, and Helios expression are shown in the new Figures 3F, 3G, and 3E, respectively. As expected, all markers associated with negative selection were highly induced in the negatively selected population VI but absent from positively selected population V cells, affirming that TCR β hiCCR7intCD69 $^-$ DN thymocytes are the products of negative selection.

The revised version fully addresses the concerns of the reviewer.

-Several conclusions such as “Examination of thymocyte maturation markers, i.e., CD5, CCR7, CD24, MHC-I, and CD69, indicated that 2D2 CD4SP cells were fully differentiated and comparable to WT CD4 T cells (Suppl. Fig. 1A)” and “Thus, the generation of CD4 T cells in 2D2 mice appeared to be intact despite their autoreactivity, and comparable to that of non-self-specific CD4 T cells in AND mice” are over-interpreted.

→We thank the reviewer for pointing out these issues. We have carefully re-evaluated our manuscript and rephrased these conclusions to avoid overinterpretations of the results. The changes are made in lines 109-112 where we changed the original sentence to “...surface markers that are associated with thymocyte selection and maturation, i.e., CD5, CCR7, CD24, MHC-I, and CD69, indicated that 2D2 CD4SP cells were comparable to WT CD4 T cells in their differentiation status.”. We also changed the conclusion in lines 120-122 to “Thus, the generation of CD4 T cells in 2D2 TCR transgenic mice appeared to be intact despite the autoreactivity of their TCR.”

The revised version addresses the concerns of the reviewer.

In many histograms, the MFI is not indicated and the expression is not quantified using several mice in several independent experiments.

→In response to the reviewer’s request, we added the MFI to the histograms and quantified the expression. Specifically, we added the MFIs to the following figures; revised Figures 1J and 1I, new Figures 5H, 5I, and 5J, and revised Figure 5K, and the results were quantified in the associated bar graphs.

Addressed

In Suppl. Fig. 1A, the authors should analyze the maturation status of 2D2 CD4 $^+$ SP thymocytes based on CD69/MHCI expression (PMID: 27043411).

→This is an excellent point. As suggested, we employed the distinct expression of CD69 versus MHC-I among CCR7+TCR β + post-selection thymocytes to analyze the maturation status of 2D2 CD4SP thymocytes. The results are added as new Suppl. Figure 3A, and they demonstrate that 2D2 CD4SP cells are mostly comprised of fully mature T cells.

Addressed

- I strongly encourage the authors to show all the data indicated as “not shown” and in particular those related to the following sentence “we next assessed the expression of a series of developmental and differentiation markers by reanalyzing public RNAseq data sets of agonistic signaled thymocytes and screening custom-made panels of antibody arrays for surface molecules (data not shown)” justifying the analysis of CD103 in negative selection.

→As suggested, we removed all “data not shown” from the revised manuscript, such that now we only explain and discuss results displayed in the manuscript. In this regard, we added the

results from screening surface molecules that are induced on negatively selected TCR β ^{hi} DN but not on positively selected TCR β ^{hi} CD4SP cells in 2D2 mice to the revision (new Suppl Figures 1D and 1E). These molecules include a series of cytokine receptors, activation markers, and integrins, and it is through this screening process that we identified the marked upregulation of integrin CD103 in TCR β ^{hi} DN thymocytes (revised Figure 1I).

Addressed

-The interpretation of results shown in Fig 4C should be reinforced by analyzing the localization of mTEC in these transgenic mice. It seems that the cortico-medullary demarcation is totally blurry in 2D2.CD103Tg β 7Tg. The authors should quantify medullary areas in 2D2 and 2D2.CD103Tg β 7Tg mice.

→The reviewer is correct to point out the blurred cortico-medullary junction (CMJ) in 2D2 mice, which we found to be further exacerbated in 2D2.CD103Tg β 7Tg mice. By retaking the H&E histochemistry images (revised Figure 6A), we affirm that the blurriness is not due to low resolution of the images but due to fragmentation of the medulla and dispersion of the CMJ. For an independent evaluation of our findings, we initiated a new collaboration with Dr. Yousuke Takahama (National Cancer Institute, NIH), who helped us to map the mTEC and cTEC locations by immunohistochemistry, affirming the altered thymic architecture in 2D2 TCR transgenic mice (new Figure 6C). Also, as requested, we quantified the medullary areas of the 2D2 and 2D2.CD103Tg β 7Tg mice, for which we employed the help of Dr. Michael Kruhlak (CCR Confocal Microscopy facility, NCI, NIH). We quantified the medullary and cortical areas using Arivis Pro software (Zeiss) and then applied machine learning pixel classifiers (random forest algorithm) to determine the ratio of medulla to cortex. The results revealed a dramatically reduced medulla to cortex ratio in 2D2.CD103Tg β 7Tg mice, demonstrating the detrimental effect of CD103 overexpression in thymic development (new Figure 6D and Materials and Methods).

Addressed

-The CD4/CD8 profile of 2D2 mice is not consistent in panels Fig. 1a and Fig.2d.

→We acknowledge that the CD4 versus CD8 profiles of 2D2 mice were inconsistent, and we have replaced them with correctly representative data, as shown in revised Figure 1A and Figure 2D.

Addressed

-Consider using an alternative approach to the t-tests that are used for some comparisons of more than two groups

→We appreciate the suggestion. Following the reviewer's suggestion, we tested the statistical power using one-way ANOVA for the Figures 1B, 1C, 1G, Figures 2C, 2D, 2E, Figures 3A, 3C, and Figures 5C, 5E, 5G, 5H, 5I, 5K and Figures 7A, 7B, 7C, and 7D, in the revision

Addressed

- statistics between WT and 2D2 are missing (Fig. 1B, C).

→As requested, we added the statistics of cell numbers and frequencies between WT and 2D2 mice to the bar graphs in the revised Figures 1B and 1C.

Addressed